# mRNA 3'UTR lengthening by alternative polyadenylation attenuates inflammatory responses and correlates with virulence of Influenza A virus

Valter Bergant [1,2], Daniel Schnepf [3,4], Niklas de Andrade Krätzig [5,6], Philipp Hubel[2], Christian Urban [1,2], Thomas Engleitner[5,6], Ronald Dijkman[7,8,9], Bernhard Ryffel [10,11], Katja Steiger [12], Percy A. Knolle [13], Georg Kochs [3,14], Roland Rad [5,6,15], Peter Staeheli[3] & Andreas Pichlmair [1,2,16] ✉

Changes of mRNA 3'UTRs by alternative polyadenylation (APA) have been associated to numerous pathologies, but the mechanisms and consequences often remain enigmatic. By combining transcriptomics, proteomics and recombinant viruses we show that all tested strains of IAV, including A/PR/8/34(H1N1) (PR8) and A/Cal/07/2009 (H1N1) (Cal09), cause APA. We mapped the effect to the highly conserved glycine residue at position 184 (G184) of the viral non-structural protein 1 (NS1). Unbiased mass spectrometry-based analyses indicate that NS1 causes APA by perturbing the function of CPSF4 and that this function is unrelated to virus-induced transcriptional shutoff. Accordingly, IAV strain PR8, expressing an NS1 variant with weak CPSF binding, does not induce host shutoff but only APA. However, recombinant IAV (PR8) expressing NS1(G184R) lacks binding to CPSF4 and thereby also the ability to cause APA. Functionally, the impaired ability to induce APA leads to an increased inflammatory cytokine production and an attenuated phenotype in a mouse infection model. Investigating diverse viral infection models showed that APA induction is a frequent ability of many pathogens. Collectively, we propose that targeting of the CPSF complex, leading to widespread alternative polyadenylation of host transcripts, constitutes a general immunevasion mechanism employed by a variety of pathogenic viruses.

Influenza A viruses (IAV) are segmented negative stranded RNA viruses belonging to the *Orthomyxoviridae* family. Two currently circulating subtypes, A(H1N1)pdm09 and A(H3N2), are responsible for the majority of annual influenza virus epidemics causing 5 million cases of severe illness leading to up to 650.000 deaths per year worldwide. Synergistic activation of innate and adaptive immune responses is required to prevent severe disease and clear the viral infection.

The best characterized immune-modulating factor of IAV is the non-structural protein 1 (NS1)[1]. NS1 exerts various strain- and species-specific functions counteracting the host's immune responses[2,3], which is in line with the concept that viral targeting of the host's defenses on several levels is beneficial for the virus[4–6]. NS1 contains two major structural domains, the N-terminal RNA-binding domain (RBD), and the C-terminal effector domain (ED). The NS1-RBD is best described for its ability to inhibit innate immunity mediated signaling cascades

through limiting the detection of viral RNA (vRNA) by pattern recognition receptors (PRRs) such as RIG-I[7–10], protein kinase R[11,12] and oligoadenylate synthetase[13]. Mechanistically, this activity is mediated through direct NS1-PRR or NS1-vRNA interactions, preventing the detection of vRNA. The NS1-ED serves as an interaction domain with multiple functions including inhibition of mRNA export through binding of the mRNA export receptor NXF1-NXT1[14], activation of PI3K, and direct inhibition of PKR[2]. In addition, most IAV strains encode for NS1 that potently interacts with cleavage and polyadenylation specificity factor 4 (CPSF4, CPSF30), which results in reduced mRNA maturation and thereby blunts cellular antiviral responses[15,16]. Previous structural characterization showed that this interaction is dependent on two distinct motifs in the NS1-ED - amino acid residues surrounding positions F103/M106 and G184[16,17]. Of note, some widely studied virus isolates, such as A/Puerto Rico/8/1934 H1N1 (PR8) and the pandemic A/California/07/2009 H1N1 (Cal09), were proposed not to interact with CPSF4 due to amino acid substitutions within NS1 at positions 103/106 and 108/125/189, respectively[5,18].

RNA polymerase II (RNAPII) activity, as well as mRNA production and maturation in general, are required for most cellular responses including the host's antiviral defense. For this reason, many viruses besides IAV target this fundamental process to facilitate virus replication and transmission. Examples include orthomyxoviruses (e.g. Thogotovirus (THOV) targets TFIIB[19], bunyaviruses (e.g. Rift valley fever virus (RVFV) depletes TFIIH[20], La Crosse virus (LACV) degrades POLR2A[21]), togaviruses (e.g. Semliki forest virus (SFV) also degrades POLR2A[22]), picornaviruses (e.g. enterovirus 71 degrades CSTF2[23]) and herpesviruses (e.g. Herpes simplex virus 1 (HSV-1) targets the CPSF complex[24]). It was previously observed that IAV infection leads to RNAPII depletion from gene bodies, transcription downstream of physiological gene transcripts and chromatin rearrangement in the intergenic regions[25,26]. However, the underlying molecular mechanisms causing these phenomena and their potential involvement in immunopathology and disease progression remains incompletely understood.

Post-transcriptional mRNA processing is absolutely essential for correct gene expression. Post-transcriptional changes, including alternative splicing (AS) and alternative polyadenylation (APA), were previously associated with a large number of pathologies[27–29], including a number of virus infections[30–32]. The role of AS was recently specifically explored in the context of IAV infection[33,34]. IAV infection also results in formation of chromatin-bound "downstream of gene" transcripts (DoGs), which are considered to be polyadenylated[25,35,36], but the mechanism of their formation and their role in the virus infection are not well understood. polyA site selection can be modulated by a variety of proteins, including the cleavage and polyadenylation specificity factor complex and a variety of RNA-binding proteins such as ELAVL1[37] (HuR) and hnRNPC[38]. Notably, APA not only influences gene expression patterns[27,39,40], but can also impact and direct mRNA and protein localization[41,42]. Efficient targeting of post-transcriptional events by small molecules or antisense oligonucleotides was recently demonstrated[43–46], highlighting mRNA processing as potential therapeutic target.

Using deep RNA sequencing (RNAseq), we show that IAV strain PR8 causes major changes of the polyadenylation landscape in human cells. Using quantitative real-time PCR (RT-qPCR)-validated APA events, we show that all tested strains of IAV, with an exception of a deletion variant lacking NS1, perturb the polyadenylation site usage in infected cells. Unbiased affinity purification followed by mass spectrometry-based analysis indicated that, surprisingly, the NS1-ED of all tested IAV strains cause APA by disturbing the function of the CPSF complex. NS1 binding to the CPSF complex critically relies on a single glycine residue at position 184 (G184) that is conserved in all IAV strains sequenced to date. The enrichment of the CPSF complex in NS1 immunoprecipitations is fine-tuned by residues at and around

positions 103 and 106, which vary among clinically relevant IAV strains and adjust the degree of transcriptional shutoff during infection. High enrichment of CPSF4 seemingly favors insurmountable RNAPII termination defects, whereas low enrichment leads to a widespread production of mature mRNAs containing elongated 3'UTRs. Notably, the ability to cause APA correlates with inhibition of inflammatory cytokine induction, including IL-6 and IFN-γ in vitro, ex vivo and in lungs of infected mice. Functionally, this low CPSF enrichment correlates with viral pathogenicity in a mouse IAV infection model. Collectively, our data demonstrate that the benefits of pharmaceutical targeting of the CPSF interaction surface of NS1 are highly relevant for all IAV strains including the 2009 pandemic strain, and thus contribute to mechanistic characterization of this potentially pan-IAV targeting strategy.

## Results

### IAV causes extensive APA of host transcripts

To evaluate IAV-induced transcriptional and post-transcriptional changes to the host transcriptome, we performed a deep transcriptomics study of the human lung carcinoma cell line A549, infected for 24 h with IAV strain A/Puerto Rico/8/1934 H1N1 (PR8). Compared to mock-infected cells we observed 777 and 555 significantly up- and down-regulated genes, respectively (Fig. 1a, Supplementary Fig. 1a, Supplementary Data 1). Moreover, analysis of alternative polyadenylation (APA) of host transcripts[47] revealed notable changes, with 3'UTRs of 119 and 18 host genes significantly lengthened or shortened, respectively (Fig. 1a, Supplementary Fig. 1a, b, Supplementary Data 2). mRNA isoform usage analysis[48,49] identified only 12 and 20 significant alternative splicing events, occurring in 3'UTRs and coding regions (Fig. 1a, Supplementary Data 3). Interestingly, genes affected by APA upon IAV infection significantly overlap with genes reported to be APA-ed across seven tumor types from tumor-normal tissue pairs in the TCGA dataset[28] (Supplementary Fig. 1c), suggesting that the IAV-induced APA might correlate mechanistically and/or functionally with APA occurring in cancer. This notion was further supported by the identification of intron- and 3'UTR-located APA events in cancer-related genes, such as TP53, HEXIM1 and TFAP2A (Supplementary Fig. 1d, e). Based on this transcriptome-wide analysis, we selected a subset of APA genes as reference markers for IAV-induced APA (Supplementary Fig. 1d). In order to test for the APA of specific genes, we implemented an RT-qPCR based test to quantify the usage of polyadenylation sites (Fig. 1b). To visualize thus acquired measurements, we use percentage of distal polyA site usage index (PDUI, log2, higher values indicate elongated transcripts) and the difference between PDUIs of treated (e.g. infected) versus untreated (e.g. mock) samples (ΔPDUI, log2, higher values indicate more elongated transcripts relative to control). To investigate if induction of APA is a conserved function of IAV, we analyzed APA fingerprints of various strains of IAV. Towards this, we infected A549 cells with IAV strains PR8, A/WSN/1933 H1N1 (WSN), A/seal/Mass/1-SC35M/1980 H7N7 (SC35M)[50], and SC35MΔNS1[51], or left them uninfected (mock), and performed RT-qPCR based quantification of APA at multiple times post infection. We observed that infection with all tested wild-type strains of IAV recapitulated APA of selected (Fig. 1c), but not unrelated (Supplementary Fig. 2a) genes. A notable exception was the NS1-deleted strain SC35MΔNS1 (Fig. 1c), which did not show any signs of APA, indicating that the NS1 protein may be responsible for the APA occurring in IAV infected cells. The comparably high abundance of viral transcripts in all conditions indicated that APA was not simply due to reduced infection rates of SC35MΔNS1 (Supplementary Fig. 2a).

To further corroborate our NGS-based findings and to test whether the APA transcripts are functionally polyadenylated and exported to the cytoplasm, we performed cellular fractionation of mock- or IAV-infected cells at 24 h post infection, followed by RNA isolation from total, nuclear and cytoplasmic fractions (Supplementary Fig. 2b). RNA integrity was evaluated by agarose gel electrophoresis and uniform

fractionation of HIST3H3 protein and *U6* snRNA in the nuclear fractions indicated successful fractionation (Supplementary Fig. 2c). For evaluation of differential 3′UTR usage, reverse transcription was performed using $(dT)_{18}$ primer (Supplementary Fig. 2b). In line with our previous findings, we observed significant elongation of transcripts' 3′UTRs in both nuclear and cytoplasmic fractions of cells infected with wild-type but not ΔNS1 mutant virus (Fig. 1d, Supplementary Fig. 2d). Interestingly, for all measured transcripts the nuclear fractions in all conditions, including uninfected, contained on average mRNA with longer 3′UTRs compared to the cytoplasmic fractions. Whether this is limited to observed transcripts or represents a more general phenomenon has yet to be explored. While these experiments do not exclude the presence of abortive transcripts, these results showed that the mRNA affected by IAV-induced APA are properly polyadenylated and exported into the cytoplasm.

To exclude that IAV-induced APA is cell line specific, we performed corresponding infection experiments using the human cell line HEK293T, resulting in similar findings in regard to elongation of host transcripts (Supplementary Fig. 2e). Moreover, we investigated APA of host transcripts upon infection of primary human tracheobronchial airway epithelial cells (hTAECs), a widely used ex vivo infection model for IAV[52]. Upon infection of hTAECs with strains PR8 and A/California/07/2009 H1N1 (Cal09), which do not induce host shutoff[18], we again detected elongation of host transcripts (Fig. 1e). Additionally, infection of immortalized mouse embryonic fibroblasts (MEFs) with IAV strain PR8 reproduced APA as observed for homologous human transcripts indicating that the observed APA in IAV infected cells is not species specific (Supplementary Fig. 2f).

Collectively, we showed that IAV infection-mediated APA is a conserved phenomenon for all tested IAV strains with the exception of a strain deleted for viral protein NS1. APA of host transcripts occurred in human and murine cell lines as well as primary human airway epithelial cells infected with different IAV strains.

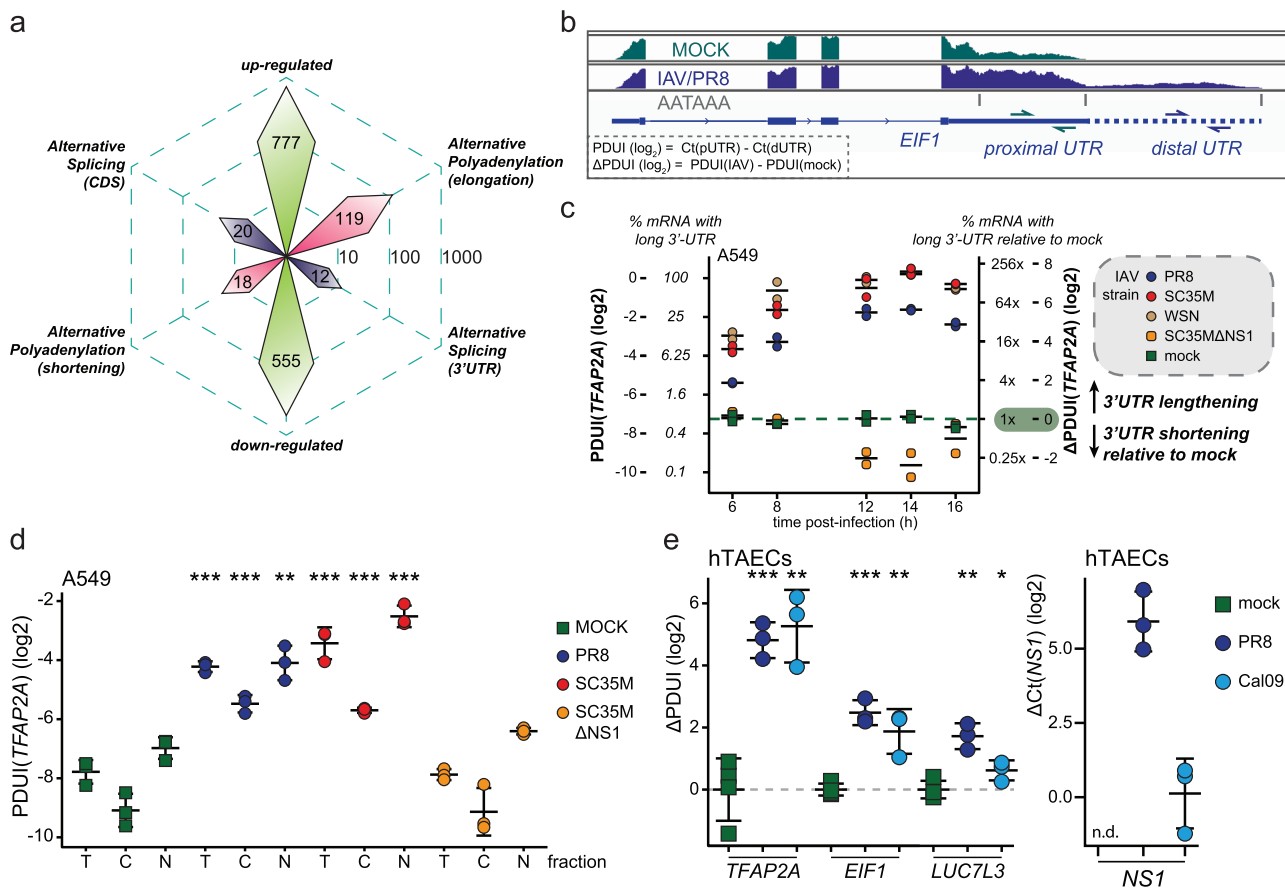

**Fig. 1 | Influenza A causes widespread alternative polyadenylation of host transcripts. a** Overview of transcriptional and post-transcriptional deregulation of host-transcripts, induced by infection of A549 cells with IAV strain PR8 for 24 h at MOI 3. Numbers of significant events are shown; related data is presented in Supplementary Fig. 1a, b. **b** Schematic read coverage across an exemplar alternatively polyadenylated human gene EIF1 with location of RT-qPCR primers used for quantification of APA shown. Calculation of PDUI and ΔPDUI from Ct values is also depicted. Further examples are shown in Supplementary Fig. 1d, e. **c** A549 cells were infected with indicated strains of IAV at MOI 3 or left uninfected (mock) and harvested at indicated times post infection. PDUI and ΔPDUI are shown as a measure of APA alongside means for 2 separately infected wells. Dashed line corresponds to uninfected polyadenylation status. Additional related data is presented in Supplementary Fig. 2a. The presented data is representative of 2 independent repeats. **d** A549 cells were infected with indicated strains of IAV at MOI 3 or left uninfected for 24 h and subjected to nucleocytoplasmic fractionation (Supplementary

Fig. 2b–d). Fractioned RNA was reverse transcribed using $(dT)_{18}$ and used for RT-qPCR based quantification of APA. PDUI as a measure of APA status is shown alongside mean +/− sd for 3 separately infected wells and is representative of 2 independent repeats. Statistics refer to comparisons between indicated samples and fraction-matched mock controls. **e** Human tracheobronchial airway epithelial cells (hTAECs) were infected with indicated strains of IAV (10e5 pfu (PR8) or 10e5 TCID50 (Cal09)) per 24-well) for 72 h and used for RT-qPCR based quantification of APA of selected genes alongside measurement of *NS1* mRNA levels. ΔPDUI as a measure of APA status is shown alongside mean +/− sd for 3 (4 mock) separately infected wells. Statistics refer to comparisons between indicated samples and gene-matched mock controls. Statistics were calculated using two-sided equal variance *t*-test. ΔCt values were calculated relative to the housekeeping gene *RPLPO*. \**p* < 0.05, \*\**p* < 0.01, \*\*\**p* < 0.001. pUTR proximal UTR, dUTR distal UTR, fraction T total, C cytoplasmic, N nuclear.

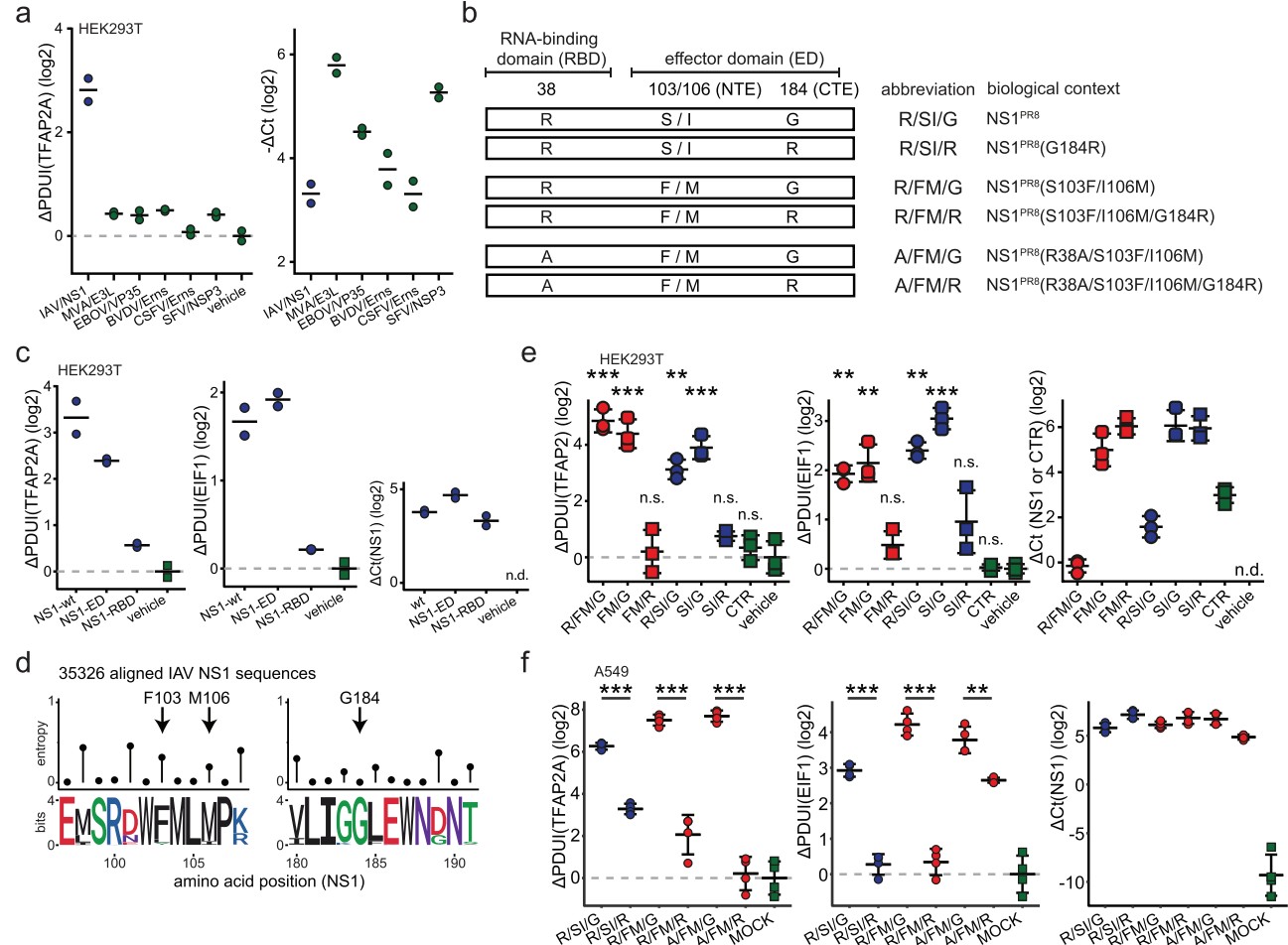

**Fig. 2 | Alternative polyadenylation of host transcripts is mediated by viral protein NS1 irrespective of amino-acids at positions 103/106 and abrogated by G184R mutation. a** Depicted viral proteins were transiently expressed in HEK293T cells via expression plasmid transfection (or transfection reagent only, vehicle), followed by quantification of APA of selected genes. ΔPDUI as a measure of APA status (left) and transgene transcript abundances (right) for 2 separately transfected wells are shown. **b** Schematic representation of previously described naming convention[5] used in respect to NS1 mutants used in this study. **c** Transient expression of full length NS1 (NS1-wt, aa 1–230), NS1 effector domain (ED, aa 74–230), NS1 RNA-binding domain (RBD, aa 1–113) or transfection reagent only (vehicle) via expression plasmid transfection, followed by quantification of expression levels and APA of selected genes. ΔPDUI as a measure of APA status and *NS1* transcript abundances are shown alongside mean for 2 separately transfected wells. **d** Amino acid sequences of IAV NS1 proteins (*N* = 35,326, NCBI Influenza virus database)), were aligned using Clustal Omega algorithm[90]. Normalized entropy (22-letter) as a measure of positional variation at indicated

positions is depicted alongside sequence logos. **e** Indicated NS1 mutant proteins or controls (CTR – Thogotovirus M, vehicle – transfection reagent only) were transiently expressed in HEK293T cells via expression plasmid transfection, followed by quantification of transgene expression levels and APA of selected genes. ΔPDUI as a measure of APA status and transgene transcript abundances are shown alongside mean +/− sd for 3 separately transfected wells. Statistics refer to comparisons between indicated samples and gene-matched vehicle controls. **f** A549 cells were infected with indicated strains of IAV (based on PR8 background, MOI 3) for 24 h, followed by quantification of *NS1* expression levels and APA of selected genes. ΔPDUI as a measure of APA status and *NS1* transcript abundances are shown alongside mean +/− sd for 4 separately infected wells. Statistics refer to comparisons between indicated samples and gene-matched mock controls. Statistics were calculated using two-sided equal variance *t*-test. n.s. *p* > 0.05, *\*p* < 0.05, *\*\*p* < 0.01, *\*\*\*p* < 0.001. ΔCt values were calculated relative to the housekeeping gene *RPLPO*.

## APA of host transcript is dependent on the C-terminal CPSF4-binding epitope of NS1

Given that NS1-deleted IAV failed to induce APA, we hypothesized that the expression of NS1 may be sufficient to reproduce APA in the absence of viral infection. We therefore transfected an expression plasmid encoding PR8-derived NS1 into HEK293T cells and quantified APA by qPCR, taking along multiple other viral dsRNA binding proteins as additional controls. We found that expression of NS1, by itself, was sufficient to reproduce the APA of selected transcripts as observed in IAV infection (Fig. 2a). To narrow down which domain of NS1 is required for this, we individually expressed the full-length NS1 protein as well as its two domains, the RNA binding (RBD) and the effector (ED) domains (Fig. 2b). Notably, expression of the ED, but not of RBD, was sufficient to induce APA of host transcripts (Fig. 2c).

The molecular mechanisms of innate immune response inhibition by NS1 are strain-dependent, and its robustness is achieved by concerted activity of both the RNA-binding and the effector domains[2]. Most strains rely on disabling the activity of the CPSF complex by binding the CPSF4, thus causing transcriptional shutoff[5,15,18]. CPSF4 is one of the central proteins mediating polyA site selection[53], and binding of NS1 to it could potentially explain the APA observed in IAV. However, some of the strains that induce APA, but not transcriptional host shutoff, namely PR8 and Cal09, are considered not to interact with CPSF complex[5,18]. This interaction is facilitated by a bipartite motif on the effector domain composed of residues 103 and 106 and their surroundings, and the residue 184[17]. The presence of amino acid residues F103 and M106 is required, but not sufficient, for host-cell shutoff caused by CPSF4 sequestration[5,18]. Interestingly, while amino acid

residues at positions 103 and 106 vary substantially between IAV strains, the G184 residue is highly conserved (Fig. 2d).

To test the potential involvement of the CPSF-binding motifs of NS1 in IAV-induced APA, we constructed expression plasmids encoding full length PR8-derived NS1 or variants harboring mutations in N- and C-terminal CPSF4-binding motifs (Fig. 2b) following previously established naming scheme[5]. Strikingly, upon transfection of these ED variants we observed that the occurrence of APA was independent of the amino acids at positions 103 and 106 (Fig. 2e). Notably, however, the occurrence of APA required the C-terminal CPSF-binding residue G184. Interestingly, the molecular relevance of this residue in IAV strain PR8 is currently unknown, but was shown to determine virulence in vivo[5]. To further corroborate results from the transfection based system, we used recombinant PR8-based virus mutants[5] to infect A549 cells. In line with results obtained by delivering isolated NS1 constructs in trans, induction of APA was dependent on the glycine at position 184 but independent of the amino acids at positions 103 and 106 (Fig. 2f).

Taken together, APA caused by IAV can be recapitulated by expression of NS1 in the absence of virus infection. The induction of APA does not depend on the presence of specific amino acids at positions 103/106 of NS1, which were previously considered to be key residues dictating the NS1-CPSF4 interaction. However, the ED of NS1, specifically G184, is required for induction of APA. This was highly unexpected because this epitope was previously not associated to any virus-host interactions in the IAV strain PR8.

## NS1 of all tested influenza A virus strains interact with the CPSF complex

In order to identify G184-dependent interactions of NS1 and thus the potential molecular reason of APA of host transcripts, we performed affinity purification coupled to liquid chromatography and tandem mass spectrometry (AP-LC-MS/MS) of exogenously (plasmid) expressed HA-tagged NS1 proteins and their effector domains in human cells (Fig. 3a, Supplementary Data 4). Based on enrichment of potential interactors of full length and ED-only constructs, we divided the interactors into ED-specific and RBD-dependent interactors for constructs with distinct N-terminal CPSF4-binding epitopes at positions 103/106 (Fig. 3a). Among the RBD-dependent interactors were many dsRNA-binding proteins, which were highly conserved between NS1 proteins with different amino acid residues at positions 103 and 106 (Fig. 3b). In stark contrast, we observe major differences in composition of ED-specific interactors between NS1 proteins, indicating that amino acids at positon 103/106 act as major determinants of ED molecular activity (Fig. 3b). Unexpectedly, we observed significant enrichment of the CPSF complex components in AP-LC-MS/MS of NS1 proteins regardless of the amino acid composition at positions 103 and 106 (Fig. 3c, Supplementary Fig. 3a). However, enrichment of CPSF complex components was substantially higher for constructs containing F/M as opposed to S/I residues at positions 103/106, suggesting that residues at these positions regulate affinity for the CPSF complex (Fig. 3c, Supplementary Fig. 3a). Further analysis revealed that CPSF4 is the only ED interactor significant changing binding specificity in line with the ability of NS1 to induce APA of host transcripts (Fig. 3a, Supplementary Fig. 3b). In order to corroborate the MS-based findings, we co-transfected expression plasmids encoding GFP-tagged CPSF4 and HA-tagged NS1 (PR8) with or without the G184R mutation in HEK293T cells and performed anti-HA immunoprecipitations. CPSF4 co-precipitated with wild-type NS1 from PR8, but not with the same NS1 harboring a G184R mutation (Fig. 3d). In sum, these findings show that NS1 from IAV strain PR8 retains some affinity towards the CPSF complex, which may be driving the observed APA of host transcripts but is insufficient to cause host shutoff.

In order to functionally test the involvement of CPSF4 in IAV-induced APA, we performed depletion of CPSF4 in human and mouse cell lines using siRNA and shRNA, respectively. In line with expected

results that CPSF4 is critical for proper cleavage and polyadenylation[25,54], we show that CPSF4 depletion recapitulates lengthening of the hallmark transcripts that are lengthened in an NS1-dependent manner upon IAV infection (Fig. 3e, Supplementary Fig. 3c). These observations are supportive of disturbance of CPSF4 by NS1 being at the center of the molecular mechanism leading to APA of host transcripts during IAV infection.

Contrary to our previous understanding of IAV, we show that the NS1 proteins from various influenza A strains are able to interact with the CPSF complex with different affinities tuned by amino acids at position 103/106, and absolutely requiring the amino acid residue G at position 184. APA of host transcripts, occurring during IAV infection in an NS1 G184-dependent manner, can be recapitulated in an orthogonal manner by depleting CPSF4 in cell lines of both human and murine origin, further highlighting the CPSF complex as central driver of IAV-induced APA.

## IAV-induced APA of host transcripts occurs independently of viral host shutoff

APA of host transcripts and viral host shutoff caused by IAV are mechanistically related events centered on NS1-based perturbation of the CPSF complex. In order to gain better understanding of their delineation, we investigated to what degree they co-occur depending on the nature of amino acids in CPSF4-binding epitopes of NS1. Towards this, we performed full proteome analysis of the human cell line A549 infected with IAV encoding amino acid variations at positions 103/106 and 184 of NS1 (Fig. 2b). Overall, we detected over 6700 host proteins, over 6000 of which were stably quantified and included in the proteomic analysis (Supplementary Data 6). Dimensionality reduction revealed clear separation of proteome signatures based on the IAV strain used for infection, with an observed tendency for strains interacting with CPSF complex and inducing APA to cluster separately from strains harboring a G184R mutation that abrogates this activity (Fig. 4a). In line with previous evidence[5], we do not observe any differences in viral protein accumulation between strains, supportive of uniform virus infectivity in vitro (Fig. 4b). However, the effect of infection with different strains on host protein abundances, in particular when comparing the effect of the G184R mutation on various backgrounds, is markedly different. We observe only a small number of differentially expressed proteins between wild-type and G184R mutant PR8 strains (Fig. 4c). In stark contrast, upon comparison of strains harboring F/M residues in the N-terminal CPSF4-binding epitope of NS1, we observe reduced abundance of a large subset of cellular proteins (Fig. 4c, Supplementary Fig. 3d). This evidence is supportive of the hypothesis that postulates the existence of low (S103/I106/G184) and high (F103/M106/G184) affinity CPSF4-binding epitopes, with low-affinity binding being sufficient to induce APA and high-affinity binding causing host shutoff.

This notion is further corroborated by additional analyses considering protein expression on a continuous distribution. We could observe a clear global protein downregulation for all strains encoding NS1 proteins that highly enrich for CPSF4 in immunoprecipitations (Fig. 4d, e). In order to visualize the repressive activity of NS1 on the proteome, we plotted the cumulative density functions depicting the distribution of protein expression changes between different strains (Fig. 4e). As expected, we observed heavy-tailed distributions towards negative values when comparing G184R-dependent profiles from strains harboring F/M but not S/I amino acids at positions 103/106. This is indicative of global repression of protein expression resulting from transcriptional host shutoff in G184-encoding strains containing FM residues at N-terminal CPSF4-binding epitope versus their G184R mutant counterparts (Fig. 4e). Given that the main function of NS1 is regulation of cellular antiviral responses, we focused more closely on expression of interferon response related proteins in cells infected with IAV bearing different NS1 mutations (Fig. 4f). The PR8 wild type

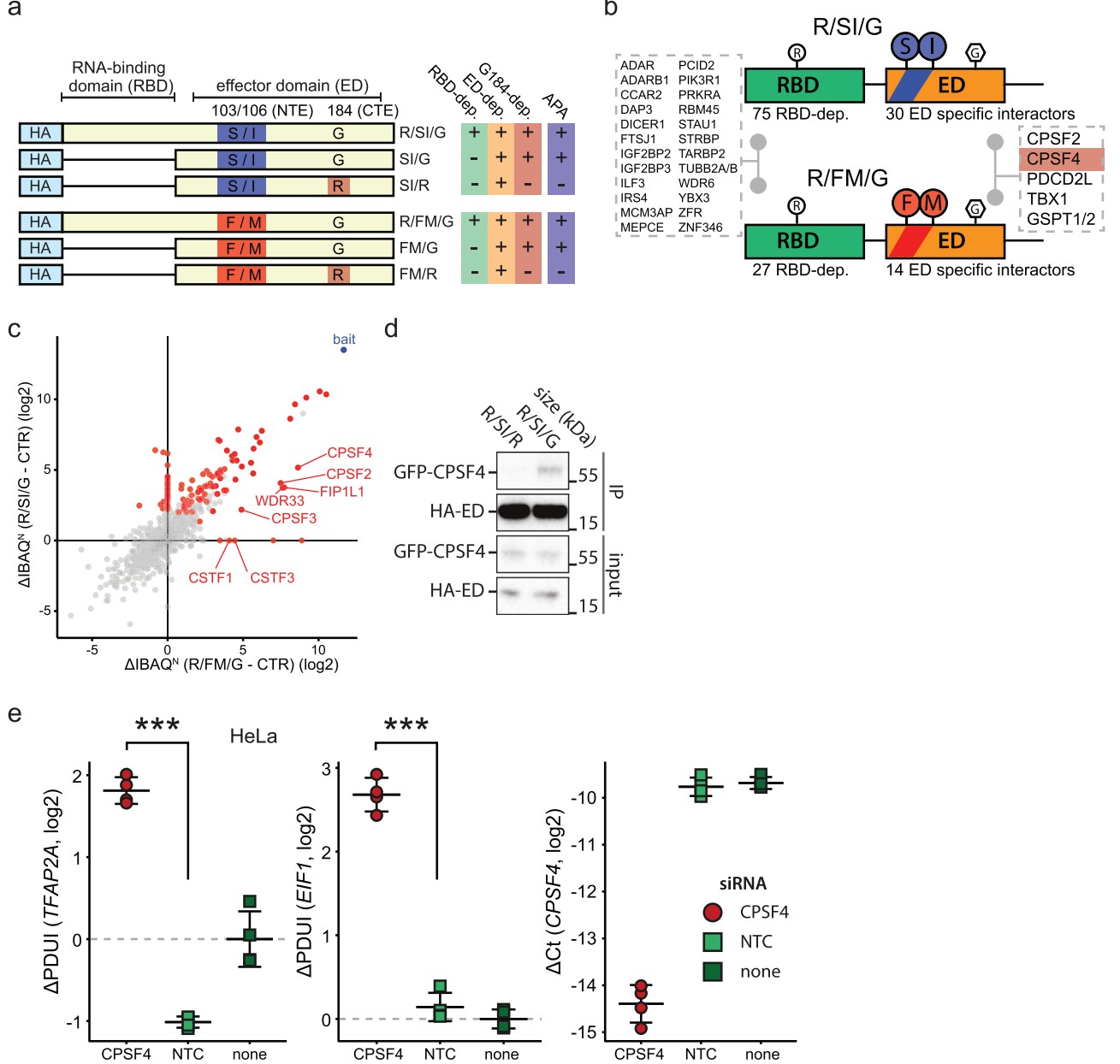

**Fig. 3 | Interaction between NS1 and the CPSF complex is diminished by variation of amino acids at positions 103/106, but only completely abrogated by variation at position 184. a** Schematic representation of HA-tagged constructs used in AP-MS experiments. **b** Schematic representation of RBD- and ED-dependent interaction partners of NS1 proteins depending on amino acids at positions 103/106. **c** Normalized IBAQ-based log2 enrichment ($\Delta IBAQ^N$) of depicted preys in comparison between R/SI/G (y-axis) and R/FG/G (x-axis) with control bait (CTR – ThoV M). Red dots show all significant interactors of the two proteins, with components of CPSF complex further highlighted. Further related data is presented in Supplementary Fig. 3a, b. **d** Western blot depicting detection of indicated proteins upon immunoprecipitation of HA-tagged R/SI/G or R/SI/R from RNase and DNase treated lysates of cells transiently expressing baits and GFP-tagged CPSF4. Depicted data are representative of 3 independent repeats. **e** HeLa cells were transfected with siRNA targeting CPSF4, non-targeting control siRNA (NTC) or transfection vehicle only (no-siRNA). 48 h post-transfection, RT-qPCR was used for quantification of expression levels of *CPSF4* and APA of selected genes. $\Delta$PDUI as a measure of APA status is shown alongside mean +/− sd for 4 separately processed wells. $\Delta$Ct values were calculated relative to the housekeeping gene *RPLP0*. Statistics were calculated using two-sided equal variance *t*-test. n.s. $p > 0.05$, *$p < 0.05$, **$p < 0.01$, ***$p < 0.001$.

strain (R/SI/G; wt, dsRNA binding, low CPSF enrichment) moderately induced expression of cellular antiviral defense proteins and this induction was similar when we used a virus with a single mutation in G184R (R/SI/R; dsRNA binding, no CPSF binding) (Fig. 4f). This is in line with the central function of the dsRNA binding domain in PR8 to suppress innate immune responses that alleviates the requirement for high affinity binding to the CPSF complex[5,55]. As expected, recombinant viruses expressing NS1 (R/FM/G; wt, dsRNA and high enrichment of CPSF4) or NS1 (A/FM/G; dsRNA binding mutant, high enrichment of

CPSF4) similarly suppressed induction of innate immune responses, which is in line with the notion that high efficiency binding of NS1 to the CPSF complex alleviates the requirement for dsRNA binding[2,5] (Fig. 4f, Supplementary Data 5). In contrast, the G184R mutation on an R/FM/ or A/FM/ NS1 background profoundly de-repressed the stringent inhibition of antiviral responses mediated by the host shutoff (Fig. 4f). These data further support the key role of 103F/106M CPSF binding motif in perturbing cellular protein expression networks and underline the importance of G184 residue in this activity. We further

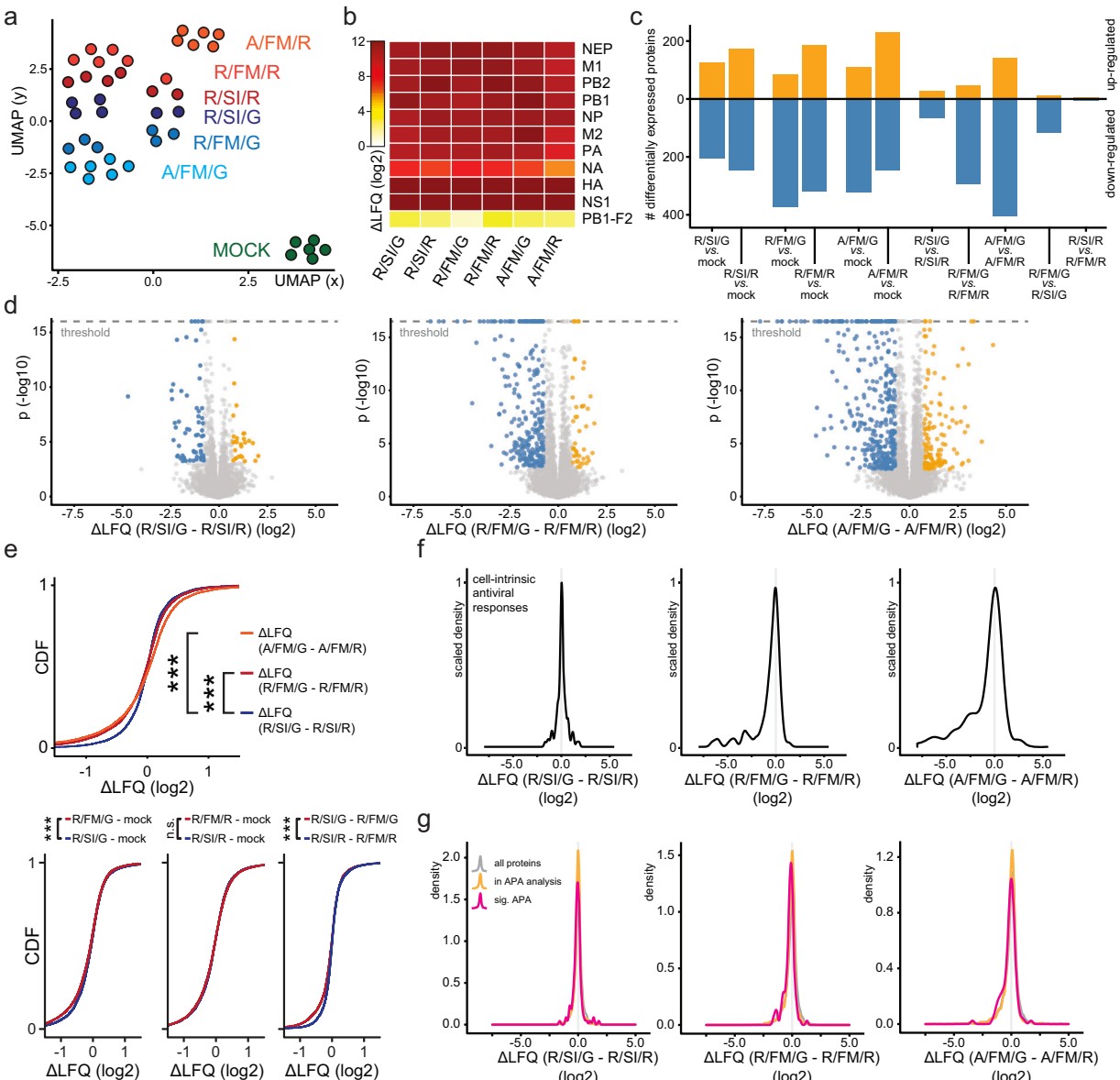

**Fig. 4 | Amino acid positions 103 and 106 dictate the occurrence of host shutoff effect but not APA. a–g** A549 cells were infected with indicated strains of IAV (MOI 3) for 24 h and used for proteome quantification using LC-MS/MS. **a** 2-dimensional representation of sample similarity, depicted through the use of uniform manifold approximation and projection (UMAP). **b** Heatmap depicting mean abundances of viral proteins upon infection with indicated viruses. **c** Plot depicting numbers of significantly up- and down-regulated proteins in comparison between indicated conditions. **d** Volcano plots showing log2 fold-changes and associated unadjusted *p*-values between indicated conditions. Statistically significant up- (orange) and down-regulated (blue) proteins (absolute log2 fold-change > 0.75, FDR-adjusted *p*-value < 0.01) are further highlighted. **e** Cumulative density function (CDF) of log2

fold-changes (displayed for clarity in limited range between −1.5 and 1.5) in comparison between indicated conditions. Statistical analysis was performed using two-sample one-sided Kolmogorov-Smirnov test comparing log2 fold-changes between individual indicated conditions with mock (e.g. R/SI/G − mock and R/SI/R − mock). **f** Plots depict scaled densities of log2 fold changes between indicated conditions for proteins belonging to the GO-term defense response to virus (GOBP:0051607). **g** Plots depict densities of protein changes between indicated conditions for all proteins (gray), proteins encoded by genes included in APA analysis (Supplementary Fig. 1b, Supplementary Data 2) (orange), and protein encoded by genes that are significantly APA in IAV infection (pink).

posed the question if APA impacts the expression patterns of proteins encoded by the affected genes (Fig. 4g). We found that the expression patterns of proteins expressed from the APA genes do not substantially deviate from the background (Fig. 4g), which strongly indicates that the APA occurring during IAV/PR8 infection does not directly hamper protein production. Presented findings support the prior observations that the host shutoff induced by IAV requires strong interaction between NS1 and the host CPSF complex and that weak interaction between NS1 and the CPSF complex is unable to cause substantial host shutoff although it is sufficient to induce perturbation in mRNA cleavage and polyadenylation patterns. Moreover, this data

indicates that APA does not have a profound effect on protein expression patterns.

Collectively, we show that while APA occurs upon infection of human cells with IAV strains regardless of amino acids at position 103/ 106, while the host shutoff, characterized by repressed expression of innate immunity related proteins as well as global repression of host protein expression in general, only occurs upon infection with strains encoding F/M amino acids at these positions. These results support the hypothesis that high efficacy binding between NS1 and CPSF4 in the context of IAV infection leads to host shutoff and APA of host transcripts, whereas low efficacy binding results exclusively in APA.

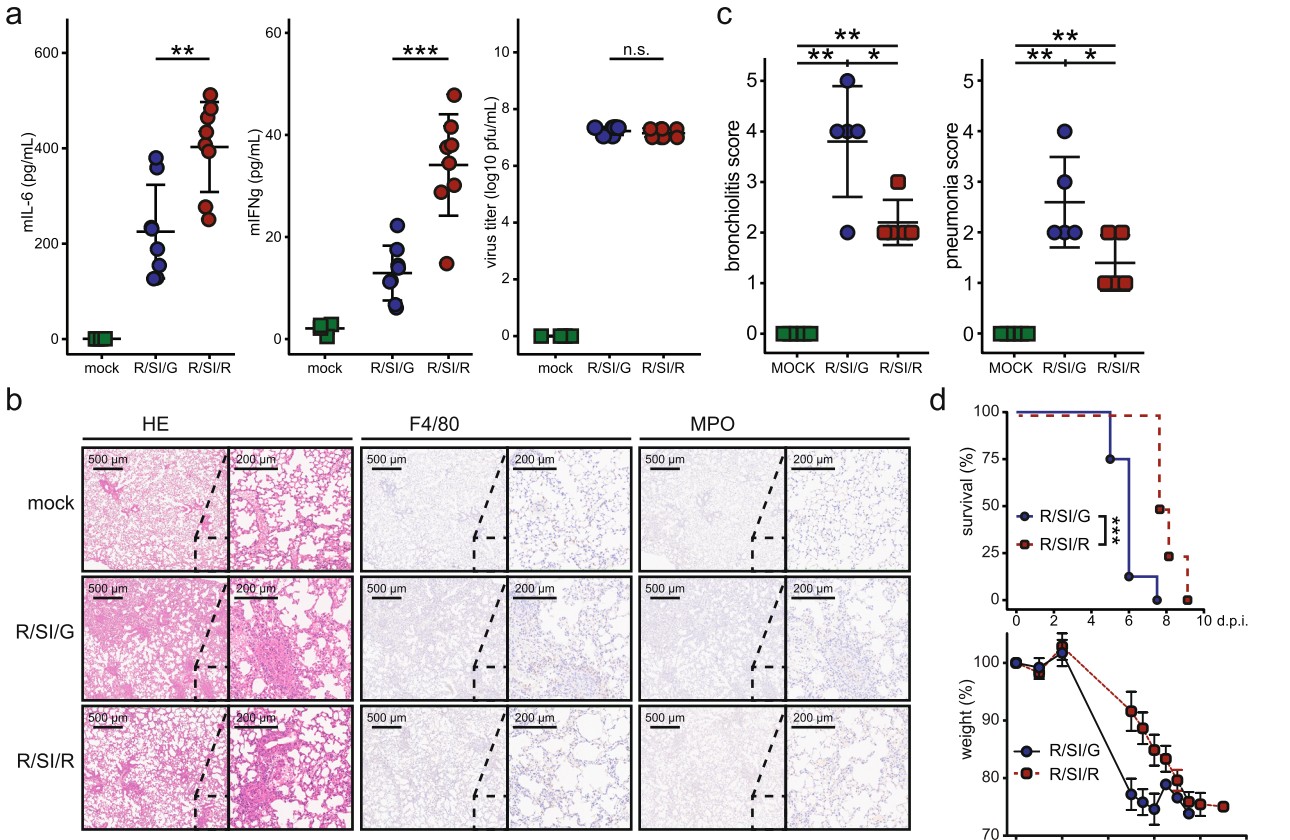

**Fig. 5 | IAV-induced APA of host transcripts inversely correlates with secretion of pro-inflammatory cytokines and interferons. a** C57BL/6J mice were infected with IAV strains R/SI/G or R/SI/R (10e5 pfu) or left uninfected. 3 days post infection we quantified secreted cytokines in BALF (left, Supplementary Data 6) and virus lung titers from lung homogenate supernatant (right). Selected significantly changing cytokines are shown alongside the virus titer; measurements from $N = 8$ (4 for mock-infection) mice are shown alongside mean +/− sd. Other significantly changing cytokines are shown in Supplementary Fig. 4a. **b** Mice were treated as in **a**, with FFPE lung samples used for histopathology evaluation. Representative images from $N = 5$ animals per condition. **c** Bronchiolitis and pneumonia scores in samples according to **b** ($N = 5$ animals per condition, shown alongside mean +/− sd). **d** $N = 8$ mice per condition were infected with indicated IAV strains and their weight followed up to day 9 post infection (mean +/− sd shown). Statistics were performed using logrank test. In **a** and **c**, statistics were calculated using two-sided equal variance $t$-test. n.s. $p > 0.05$, *$p < 0.05$, **$p < 0.01$, ***$p < 0.001$.

## Abrogation of IAV-induced APA correlates with attenuation of the virus and reduced secretion of pro-inflammatory cytokines

The G184R mutation has a pronounced effect on virulence of the PR8 virus in mice, but the underlying mechanism remained unclear[5]. Remarkably, virus replication, inhibition of the IFN response, protein localization[5] and structure[17] are not perturbed in vitro. This led us to hypothesize that the difference in virulence between the wild-type and mutant strains might result from differences in the interaction of infected primary target tissue with surrounding cell types, altering the secretion of pro- and anti-inflammatory cytokines and chemokines. To explore the pathological differences between the strains in vivo, we infected B6 mice with wild-type PR8 and G184R mutant viruses and quantified cytokines and chemokines in bronchoalveolar lavage fluid (BALF). We observed no differences in the levels of cytokines or chemokines, nor viral titers, at one day post infection (Supplementary Data 6). However, at three days post infection, we observed significantly higher levels of IL-6 and IFN-γ in BALF of G184R mutant virus-infected mice (Fig. 5a, Supplementary Fig. 4a). As expected[5], viral loads in lung homogenates did not differ between wild-type and mutant virus at 3 days post infection (Fig. 5a). To identify pathophysiological differences, we performed histological evaluation of mouse lungs at 3 days post infection (Fig. 5b). Wild-type virus as opposed to the G184R mutant caused significantly enhanced bronchiolitis and pneumonia, indicating increased severity of the pathological state (Fig. 5b, c). In line with the outcome of the histological evaluation, mice infected with

the G184R mutant virus survived the lethal infection for a significantly longer period of time (Fig. 5d).

In sum, we show that the G184R mutant strain that lacks the ability to induce APA of host transcripts has remarkably reduced virulence in vivo relative to the parental PR8 strain, accompanied by higher secretion of cytokines and reduced pathology. While the molecular mechanism connecting the G184R mutation to enhanced cytokine secretion remains unclear, the superior tissue response in lungs is correlating with a reduced virulence of the G184R strain.

## IAV-induced APA of host transcripts inversely correlates with secretion of IL-6

To corroborate the inverse correlation between IAV-induced APA of host transcripts and secretion of pro-inflammatory cytokines, we performed infection experiments with wild-type and G184R mutant viruses at different MOIs in vitro. We chose to focus on IL-6 as a widely accepted indicator of pro-inflammatory responses, which is also robustly regulated both on the level of secreted cytokine amount as well as mRNA abundance. Notably, in line with the in vivo data, infection of MEFs with the G184R mutant virus resulted in significantly higher IL-6 secretion as compared to the wild-type strain, which is also reflected at the transcript level (Fig. 6a). Moreover, we observed enhanced levels of secreted IL-6 in the human A549 cell line, infected with the G184R mutant virus at MOI 3 for 18 h (Fig. 6b). To further confirm the inverse correlation between IAV-induced APA of host

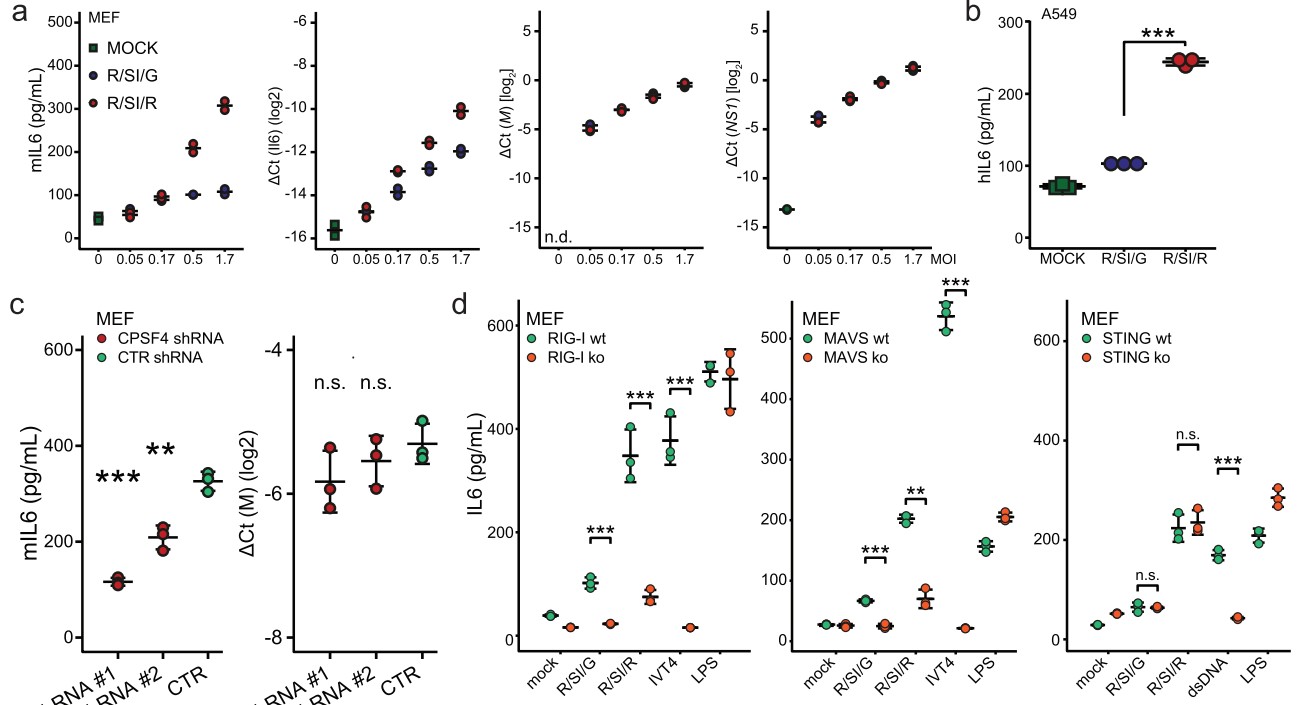

**Fig. 6 | APA-deficient IAV strain induced IL-6 secretion is RIG-I and MAVS dependent and can be circumvented by concomitant CPSF4 knock-down.**
**a** Mouse embryonic fibroblasts (MEFs) were infected with indicated strains of IAV at various MOIs (x-axis) for 24 h, followed by quantification of (left) secreted IL-6 by ELISA, *IL-6* mRNA (middle) and viral transcripts *M* and *NS1* (right) by RT-qPCR for two separately infected wells. The presented data is representative of 2 independent repeats and shown alongside mean. ΔCt values were calculated relative to the housekeeping gene *ACTB*. **b** A549 cells were infected with indicated strains of IAV at MOI 3 for 24 h, followed by quantification of secreted IL-6 by ELISA for three separately infected wells, shown here alongside mean +/− sd. The presented data is representative of 3 independent repeats. **c** MEFs (3 independent wells) were transfected with shRNA targeting Cpsf4 or Hnrnpc (CTR) 1 day prior to infection

with R/SI/R at MOI 3. 24 h post infection, secreted IL-6 was quantified by ELISA (left) and viral transcript M abundance by RT-qPCR, shown here alongside mean +/− sd. Further related data is presented in Supplementary Fig. 4b. The presented data is representative of 2 independent repeats. ΔCt values were calculated relative to the housekeeping gene *ACTB*. **d** RIG-I (left), MAVS (middle) and STING (right) KOs were used alongside respective wt controls and either infected with indicated IAV strains (MOI 3) or treated with indicated stimuli (IVT4 in vitro transcript 4, RIG-I ligand[74], LPS lipopolysaccharide, TLR4 ligand, dsDNA double-stranded DNA, STING ligand). 24-h post infection/treatment, secreted IL-6 was quantified by ELISA for 3 independently processed wells and shown alongside mean +/− sd. The presented data is representative of 2 independent repeats. Statistics were calculated using two-sided equal variance *t*-test. n.s. $p > 0.05$, *$p < 0.05$, **$p < 0.01$, ***$p < 0.001$.

transcripts and IL-6 secretion, we recapitulated APA by depletion of CPSF4 in MEFs (Supplementary Fig. 4b) using shRNA and infected them with the G184R mutant virus. CPSF4 depletion did not affect cell viability as compared to control shRNA treatment (Supplementary Fig. 4b). Notably, however, we observed a clear reduction of IL-6 secretion (Fig. 6c). These data link the activity of the CPSF complex to cytokine production and/or secretion.

To delineate which pattern recognition receptor (PRR) marks the starting point of the inflammatory cascade leading to the observed immunomodulatory phenotype, we performed infections of MEFs carrying mutations in RIG-I, MAVS or STING with the wild-type or G184R mutant virus for 24 h, followed by quantification of secreted IL-6. The increase in IL-6 secretion in response to the mutant virus was largely dependent on RIG-I and MAVS, but independent of STING (Fig. 6d). These findings strongly indicate the existence of an inverse correlation between the ability of IAV to induce APA of host transcripts and the secretion of pro-inflammatory cytokines such as IL-6 resulting from RIG-I activation upstream of the MAVS signaling cascade.

## APA of host transcripts is observed after infection of human cells with several different viruses

The striking evolutionary conservation of the NS1-CPSF interaction in IAV strains led us to hypothesize that other, unrelated viruses could also employ APA towards a similar immunomodulatory effect. To test this hypothesis, we infected the human A549 cell line with a spectrum of viruses belonging to different virus families and evaluated

polyadenylation patterns of genes affected by IAV. Notably, several viruses induced APA (Fig. 7a, Supplementary Fig. 5a). SFV and HSV-1 infections recapitulated the IAV-induced APA fingerprint, while RVFV (clone 13) and LACV, both members of the order *Bunyavirales*, caused a less pronounced APA signature. In contrast, other viruses tested, including Thogoto virus (THOV), Vesicular stomatitis virus (VSV), Zika virus (ZIKV) and Vaccinia virus (VACV), did not cause appreciable APA of the assayed transcripts.

Interestingly, HSV-1 has been shown to interact with CPSF complex through ICP27[24], leading to transcription beyond termination sites observed to be used under physiological conditions[24,56,57], which may rationalize the APA fingerprint observed herein. RVFV NSm was shown to interact with CPSF2 in a yeast two hybrid system[58], but this interaction, to the best of our knowledge, was never been tested in the context of mammalian cells. LACV causes degradation of POLR2A in a CTD phosphorylation pattern dependent manner[21] which is a shared activity with SFV, which also causes degradation of RNA polymerase II subunit POLR2A in mammalian cells[22]. However, perturbation of cleavage and polyadenylation patterns by SFV and LACV, which interfere with POLR2A stability, was highly unexpected. In order to explore the molecular pathways leading to SFV-induced APA, we performed full proteome analysis of mock- and SFV-infected human cell line THP1 (Supplementary Data 7). We stably quantified over 5000 host proteins, including RNAPII complex components and the complexes it interacts with through its CTD encoded by POLR2A. As expected, the abundance of RNAPII complex components, including POLR2A, was reduced,

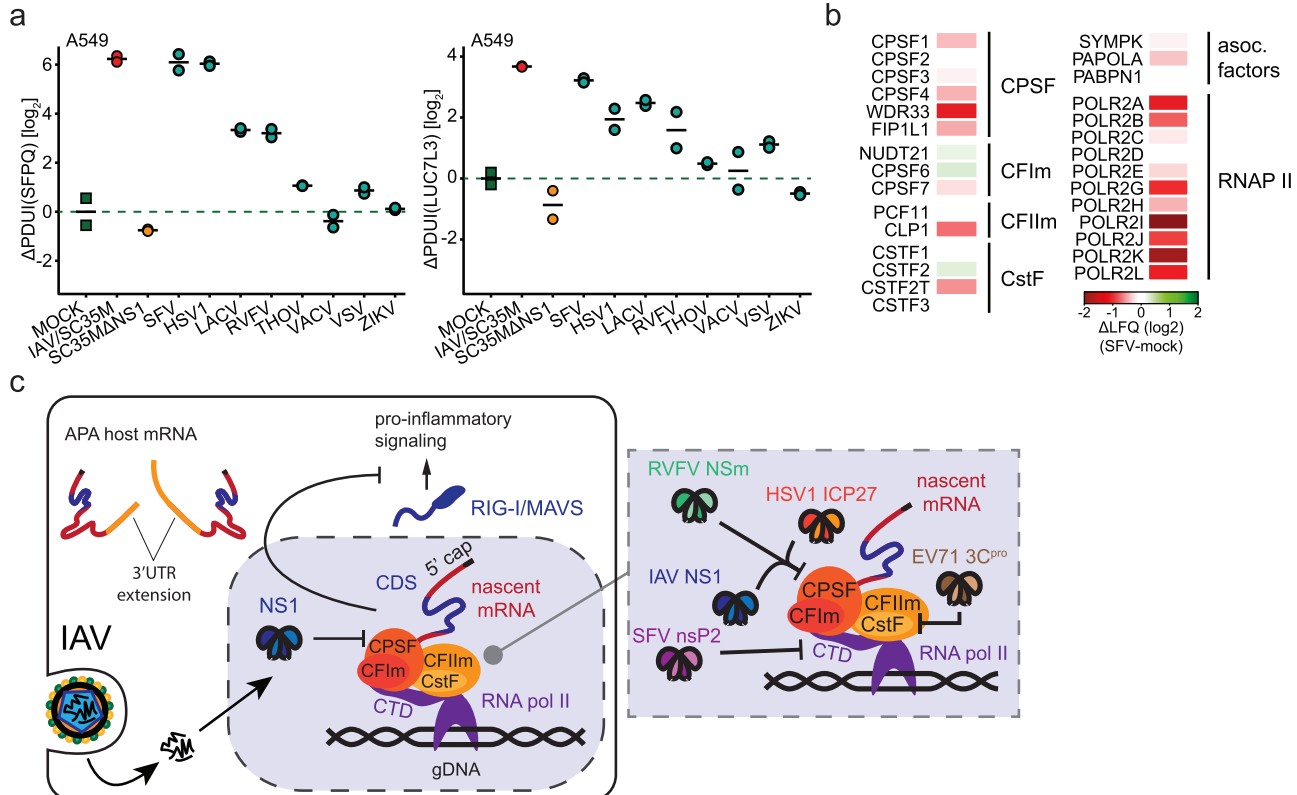

**Fig. 7 | Host transcriptional and in particular polyadenylation machinery is a common target point of pathogenic viruses. a** A549 cells were infected with indicated viruses at MOI 1 for 24 h, followed by quantification of APA of depicted host transcripts. ΔPDUI as a measure of APA status is shown alongside mean for 2 separately infected wells. Further related data is shown in Supplementary Fig. 5a. SFV Semliki Forest Virus, HSV1 Herpes Simplex Virus 1, LACV La Crosse encephalitis Virus, RVFV Rift Valley Fever Virus, THOV Thogotovirus, VACV Vaccinia virus, VSV Vesicular Stomatitis Virus, ZIKV Zika Virus. **b** THP-1 cells were left uninfected or infected with SFV at MOI 1 for 24 h, followed by LC-MS/MS based quantification of protein abundances. Protein abundance changes between infected and uninfected conditions for selected transcription-related complex components is depicted. Further related data is presented in Supplementary Fig. 5b, c. **c** Schematic representation of viral targeting strategies centered on the host transcriptional machinery and its interface with the complexes involved in cleavage and polyadenylation.

leading us to hypothesize that the degradation of POLR2A might extend to its interacting partners (Fig. 7b). Notably, we also identified a reduction in abundance of the CPSF complex components upon SFV infection as compared to mock, which was especially prominent for WDR33 (Fig. 7b, Supplementary Fig. 5b). We additionally validated these MS-based findings by WB, where we showed approximately 2-fold reduction in abundance of CPSF complex components CPSF3 and CPSF4 upon SFV infection of A549 cell line for 24 h (Supplementary Fig. 5c). It is possible that the reduction in abundance of these proteins underlies the SFV-induced APA. In summary, we demonstrate that perturbation of the cellular mRNA cleavage and polyadenylation machinery influencing polyadenylation patterns of host transcripts is a common occurrence accompanying infection of human cell lines with a wide spectrum of viruses that potentially achieve it through distinct molecular mechanisms (Fig. 7c).

## Discussion

We characterized APA patterns upon IAV infection of human cells and showed correlation of APA with inhibition of host cytokines in vitro and in vivo, providing a possible explanation for the previously observed decrease in virulence of G184R mutant PR8 strain[5]. We found that NS1 is required and sufficient to induce APA of host transcripts in the presence as well as in absence of IAV infection. We revealed that NS1 proteins of all tested IAV strains maintain either high or low enrichment of the CPSF complex in immunoprecipitation experiments. Based on the presented results we propose a hypothesis that high enrichment of the CPSF complex by NS1 leads to APA, inhibition

of the cell intrinsic innate immune responses and transcriptional host shutoff, while low CPSF enrichment causes APA but no strong inhibition of the cell intrinsic innate immune response or host shutoff. We further show that the epitope involved in modulating the NS1-CPSF4 enrichment plays a pivotal role in virus attenuation in vivo, accompanied by increased secretion of type II IFN and NF-kB-dependent cytokines.

It is currently unclear which genes or genetic elements that are affected by CPSF complex inhibition are responsible for the observed altered cytokine regulation. Moreover, an overshooting pro-inflammatory response was previously associated with severe disease progression[59]; how the altered cytokine expression patterns observed in vivo cause a less pronounced pathology and attenuate disease progression is yet to be elucidated. However, we have molecularly located this IAV-mediated inhibition and further established that the APA pattern caused by IAV infection can also be found in cells infected with other, unrelated viruses. We show that, among the viruses included in this study, APA caused by HSV-1 and SFV closely resembles that of the IAV. While one could envision similar molecular mechanisms – i.e. the CPSF complex targeted by IAV and HSV-1[24,60], there was so far no clear connection between SFV and transcriptional termination. We here provide evidence that the abundances of RNAPII CTD-associated CPSF and CFIIm complexes involved in cleavage and polyadenylation are reduced in SFV infected cells, which may point towards a conserved mechanism leading to APA.

Beyond IAV, HSV-1 and SFV, two members of bunyaviridae family, RVFV and LACV, also cause a similar yet apparently distinct APA

signature. LACV protein NSs was previously shown to cause degradation of POLR2A in a CTD phosphorylation-dependent manner. RVFV is, however, not known to directly affect the function of RNAPII, but instead perturbs the activity of general transcription factor 2 H (TFIIH) through viral protein NSs. This interaction is not relevant in the case of the RVFV strain used in this work (RVFV clone 13), which harbors a deletion in the NSs protein, widely considered to result in production of a non-functional protein. An interaction between RVFV NSm protein and CPSF2 has been observed in a yeast 2 hybrid screen[58]. However, whether the interaction takes place in the context of infection in a mammalian cell setting remains to be elucidated. Presented evidence leads us to propose that these and potentially other viruses target a susceptible part of host mRNA post-transcriptional maturation machinery that is involved in immunomodulation through a so far unknown molecular mechanism.

Long chromatin-bound downstream-of-gene transcripts[35,61] (DoGs) were previously proposed to be caused by either NS1-dependent[36] or independent[25] mechanisms. The NS1-independent mechanism was proposed to be caused by cellular stress accompanying virus infections[25], which may be hyperosmotic in nature[25,61]. Formation of DoGs during hyperosmotic shock was recently associated with loss of integrator complex from RNAPII[62]. Furthermore, hyperosmotic stress was recently shown to cause rapid sequestration of CPSF6 from the sites of active transcription into nuclear phase separated granules[63]. The phase separation of CPSF6[63] following hyperosmotic stress may in particular contribute to the high degree of correlation between DoGs previously observed in stressed and IAV-infected cells[25] and may further contribute to the hyper-inflammatory landscape of pathophysiological hyperosmotic conditions[64]. While DoGs and RNAPII occupancy loss from gene bodies, a putative hallmark of transcriptional host-shutoff, were previously recognized as separate phenomena[25], further studies are required to mechanistically and functionally delineate them. Moreover, the potential similarity between mechanisms of formation as well as biological functions of DoGs and APA transcripts in IAV infection remains to be elucidated.

Alternative polyadenylation was associated to a number of biological functions such as tissue specific protein expression and proliferative potential of cells[40,53,65,66]. It was furthermore observed in numerous diseases, such as VSV infection[32] and cancer[27,39]. Despite the association of APA with multiple pathologies, determination of causal relationships between distinct APA events and molecular disease progression is highly challenging. The main reasons for this are the limited power of detection of APA events for numerous poorly expressed transcripts (e.g. transcription factors and cytokines), limited correlation of APA with mRNA and protein expression, and the lack of methodology that would enable polyA site use modulation in follow-up studies. Another reason for this may be the complex regulatory nature of 3′UTRs. It was recently shown that mRNA 3′UTRs can not only impact its metabolism, but also mRNA and protein localization via 3′UTR-dependent mRNA sequestration into TIS-granule rich ER (TIGER) domains[41,42]. Protein translation in the TIGER domains was shown to be efficiently coupled to the plasma membrane directed transport, which provided the mechanistic basis for the mRNA 3′UTR-dependent protein localization[41,42]. However, protein localization changes and even abundance variations of scarcely expressed receptor proteins are difficult to capture on a global manner. Such changes may have a strong impact in inflammatory diseases such as virus infections, where disease course-determining processes are driven by dynamically localized proteins involved in cell-to-cell communication, para-, and autocrine signaling.

Finally, the treatment potential of preventing the induction of APA by IAV or other viruses remains to be explored. Small molecule inhibitors of IAV NS1 targeting the G184 region, and in particular the NS1-CPSF4 interaction, were previously described[67–70]. Their in vitro

phenotypes, in particular their strong ability to suppress PR8 replication in an interferon response dependent manner[68], however do not fully overlap with the expectations from behavior of G184R mutant strains as observed by us and others[5]. Collectively, our data demonstrate that the benefits of pharmaceutical targeting of the CPSF interaction surface of NS1 are highly relevant for all IAV strains including the 2009 pandemic strain, and thus contribute to mechanistic characterization of this potentially pan-IAV targeting strategy. Our data also demonstrate that the main benefits of inhibiting the NS1-CPSF interaction only become visible in vivo, however, this may be to an extent strain-dependent. Furthermore, restoring the activity of the CPSF complex may be of therapeutic benefit against a broad spectrum of viruses.

## Methods
### Cells, plasmids and reagents
HEK293T, HeLa, A549, Vero E6 cells (CRL-1586, ATCC) and MEFs[71] were grown under standard culturing conditions as described previously[71]. hTAECs were cultured as described previously[52]. For gene knock-down experiments, siRNA siGENOME reagents against human CPSF4 (M-012292-02) and non-targeting control (D-001206-14) were purchased from Horizon, PerkinElmer, and used as previously described[71]. shRNA against mouse CPSF3 (shRNA #1: CCAGCAAACCAGTGAATTTAT, shRNA #2: GCTGCATGACATACCCATTTA and shRNA#3: GCATGACA TACCCATTTACTA), CPSF4 (shRNA #1: GCCATGTCTGTCCTTTCATTT, shRNA #2: CAGATGCACCAAAGGGCATTT and shRNA #3: CAGATG CACCAAAGGGCATTT) and hnRNPC (CTR; CCCTCTACTCAGTTCCT CATT and CAGTAGAGATGAAGAATGAAA) were used as described previously[72]. MEFs with genetic ablation of MAVS and STING, alongside wild-type controls, were a gift from Michael S. Diamond and Søren Riis Paludan, respectively. Production of lentiviruses, transduction of cells and antibiotic selection were performed as described previously[73]. All cell lines were tested to be mycoplasma-free. For stimulation of cells, dsRNA (Sigma, D1626), IVT4[74] and LPS (Sigma, L2630) were used. Transfection of plasmids was performed using polyethylenimine (PEI 25 K, Polysciences) – unless otherwise stated, medium exchange was performed 4-hours- and the cells were harvested 24-h post transfection. Transfection of other nucleic acids was performed using Metafectene Pro (Biontex). Expression plasmids harboring GFP-CPSF4, viral dsRNA binding proteins and NS1 mutants were prepared in pCAGGS backbone as previously described[19]. Resazurin reduction assay as a measure of cell viability was performed as described previously[75].

### Virus strains, stock preparation and in vitro infection
Unless otherwise stated, the virus stocks were produced in Vero E6 cells as previously described[73]. IAV strains PR8, SC35M, SC35M dNS1[51], WSN and Cal09[76] were described previously[5]. Recombinant IAV strains, produced in chicken eggs, R/SI/G, R/SI/R, R/FM/G, R/FM/R, A/FM/G and A/FM/R were a gift from Georg Kochs[5]. SFV[77], HSV-1 (gift from Søren Riis Paludan), LACV and RVFV (gift from Friedemann Weber), ThoV[19], VacV MVA[78], VSV[79], ZIKV[72] were previously described.

Cells were washed with PBS prior to addition of infectious inoculum (unless otherwise stated at MOI 3) diluted in ice-cold PBS. Cells were incubated for 1 h on ice prior to removal of inoculum and warm medium replacement. At the time of sample harvest, the cells were washed once with 1x PBS buffer and lysed in LBP (Macherey-Nagel) or 1x SSB lysis buffer (62.5 mM Tris HCl pH 6.8; 2% SDS; 10% glycerol; 50 mM DTT; 0.01% bromophenol blue) for RT-qPCR or western blot analyses, respectively. The samples were heat-inactivated and frozen at −80 °C until further processing. Sampled supernatants were stored frozen at −80 °C until further processing.

### Plaque assays
Confluent monolayers of Vero E6 cells were infected with serial five-fold dilutions of virus supernatants (from 1:100 to 1:7812500)

for 1 h at 37 °C. The inoculum was removed and replaced with serum-free MEM (Gibco, Life Technologies) containing 0.5% carboxymethylcellulose (Sigma-Aldrich). Two days post infection, cells were fixed for 20 min at room temperature with formaldehyde directly added to the medium to a final concentration of 5%. Fixed cells were washed extensively with PBS before staining with H2O containing 1% crystal violet and 10% ethanol for 20 min. After rinsing with PBS, the number of plaques was counted and the virus titer was calculated.

### Cell fractionation

Nucleocytoplasmic fractionation of A549 cells was performed as previously described[80]. In short, $10^6$ cells were scraped from 10 cm dish and washed with ice-cold PBS. The cells were resuspended in 0.2 mL of Dautry buffer (10 mM Tris-HCl pH 7.5, 140 mM NaCl, 1.5 mM $MgCl_2$, 10 mM EDTA, 0.5 % NP-40, protease inhibitor cocktail cOmplete, Merck, RNase inhibitor RNasin Plus, Promega) and inclubated 3 min on ice before centrifugation for 5 min, 3.000 rpm at 4 °C. Supernatant (cytoplasmic fraction) was further cleared by centrifugation for 1 min at 15.000 g. Pellet (nuclei) was further washed twice with Dautry buffer and centrifugation for 5 min, 5.000 rpm at 4 °C, and finally lysed in SSB buffer (62.5 mM Tris HCl from 1 M stock solution with pH 6.8, 2% SDS, 10% Glycerol, 50 mM DTT and 0.01% Bromophenol Blue in distilled water) or Trizol (Thermo Fisher Scientific) for protein or RNA quantification, respectively.

### Transcriptomic analysis of IAV-infected A549 cells

To evaluate IAV-induced transcriptional and post-transcriptional changes, we performed deep RNA sequencing of A549 cells, infected for 24 h with IAV strain A/Puerto Rico/8/1934 H1N1 (PR8) at MOI 3. The single-end sequencing was performed using Illumina Next-Seq 2000 employing Illumina stranded mRNA Library Prep Kit according to manufacturer's instructions. The raw sequencing data was processed with Trimmomatic version 0.36[81]. Trimmed reads were acquired by removing Illumina TruSeq3 adapters and bases at the start and end of each read, for which the phread score was below 25. Further reads were clipped if the average quality within a sliding window of 10 fell below a phread score of 25. Conclusively reads smaller than 50 bases were removed.

For mapping, the human gene annotation release 29 and the corresponding genome (GRCh38.p12) were derived from the GENCODE homepage (https://www.gencodegenes.org/). The virus (A/Puerto Rico/8/1934, H1N1) segment annotation and reference genome were derived from the following NCBI reference sequences: NC_002016.1, NC_002017.1, NC_002018.1, NC_002019.1, NC_002020.1, NC_002021.1, NC_002022.1, and NC_2023.1. A custom genome was generated to include human as well as virus sequences and genes/segments. STAR version 2.6.1c[82] was used to map the trimmed sequencing data to the combined reference genome. STAR parameters were adapted from protocol recommendations for STAR[83] and STAR-Fusion[84]. For each alignment, the number of accepted multimappings was set to 10 and the number of accepted mismatches to 3. The maximum intron size and gap size between two read mates was set to 500000. For chimeric segments the following parameters were applied: a minimum length of 10, minimum total score of 1, maximum gap of 3 between segments, a maximum difference of scores equal to 30 and a minimum score difference of 1 between the best score and the next one. Chimeric junctions' parameters were adjusted to only accept a minimum overhang of 10 and the penalty for non-GT/AG junctions was set to 0. Further the maximum number of stitching of splice junctions was adjusted to −1 for GT/AG and CT/AC motifs and 5 for non-canonical and all other motifs. After mapping, one uninfected control sample showed a high amount of reads mapping to viral segments. This virus contaminated sample was removed in all subsequent analysis.

### Analysis of alternative polyadenylation

Reads mapped to the genome were converted from bam to bedgraph format using bedtools version 2.26.0 using default parameters. APAtrap version 1.0[47] was then used to identify events of alternative polyadenylation. The identification of distal 3'UTR regions (identifyDistal3UTR) and detection of genes with differential APA site usage between untreated and virus infected samples (predictAPA) was performed with default parameters. The following cut-offs were used to determine significance: FDR-adjusted $p$-value < 0.05 and PD > 0.2 and abs(r) >0.1. PD: percentage difference of APA site usage. r: Pearson product moment correlation coefficient, positive values indicate elongation and negative values shortening of mRNA relative to control condition

### Analysis of differential expression

DESeq2 version 1.36.0[85] was used to determine differentially regulated genes between untreated cells and virus infected cells. Genes were determined to be differentially expressed, if the absolute log2 fold change was greater than 2 and the FDR-adjusted p-value was below 0.01.

### Analysis of alternative isoform usage

The human reference transcriptome (release 29, GRCh38.p12) was derived from GENCODE. Transcript read count quantification was performed using the pseudo alignment software SALMON v0.13.1[86]. The quantification mode was set to U (single-end, unstranded) with additional correction for sequence-specific and GC bias. The Bioconductor package IsoformSwitchAnalyzeR version 1.8.0[48,49] was used to identify differential isoform usage between untreated and virus infected samples. Transcripts, quantified by SALMON, were imported using IsoformSwitchAnalyzeR. Isoform switch analysis was performed on untreated against virus infected samples with a false discovery rate (FDR) cut-off of 0.05. Additionally, a difference in isoform fraction (dIF) cut-off of 0.3 and a coding potential cut-off of 0.725 were used, as suggested in the IsoformSwitchAnalyzeR vignette for human samples.

Visualization of read coverage on distinct host genes was performed using Integrative Genomics Viewer (Broad Instiue).

### Quantitative LC-MS/MS experiments and analysis

For affinity purification (AP-MS), two confluent 15 cm dishes of HEK293T cells per quadruplicate were transfected with 40 µg of expression plasmids (NS1 R/SI/G, SI/G, SI/R, R/FM/G, FM/G and FM/R; controls: two separate sets of transfection reagent only, ThoV M[19]). The medium exchange was performed 6 h post-transfection and the cells were harvested 2 days later. Cell lysis (with sonication, 15 cycles at high setting, Bioruptor, Diagenode) and affinity purification (HA-beads, Sigma, A2095, without addition of DNases/RNases) was performed as described previously[73].

For whole-cell proteomics (FP-MS) of IAV or SFV infected cells, A549 or THP-1 cells were grown to 50% confluence in 6-well format and infected with indicated viruses at MOI 3. 24-hours post infection, the cells were washed twice with PBS and lysed in guanidinium chloride buffer (6 M guanidinium chloride, 10 mM TCEP, 40 mM chloroacetamide (CAA), 100 mM Tris-HCl pH 8), boiled at 95 °C for 8 min and sonicated (10 min, 4 °C, 30 s on, 30 s off, high setting on Bioruptor Plus, Diagenode). Protein concentrations of cleared lysates were normalized to 50 µg.

Proteins were denatured, reduced, alkylated and digested by addition of 200 µl digestion buffer (0.6 M guanidinium chloride, 1 mM tris(2-carboxyethyl)phosphine (TCEP), 4 mM CAA, 100 mM Tris-HCl pH 8, 0.5 µg LysC (WAKO Chemicals) and 0.5 µg trypsin (Promega) at 30 °C overnight. Peptide purification on StageTips with three layers of C18 Empore filter discs (3 M) and subsequent mass spectrometry analysis was performed as described previously[73]. Raw files were processed with MaxQuant[87] version 1.5.6.2 and default parameters, using

label-free quantification and match between run enabled and searched against forward and reverse sequences of the human proteome (Ensembl GRCh37-75 or Uniprot reviewed canonical isoforms) supplemented by baits (AP-MS) or viral protein (FP-MS) sequences.

The analysis of MS data sets was performed using R version 4.0.2. iBAQ (intensity based absolute quantification intensities, used for analysis of AP-MS experiments and SFV FP-MS experiment) or LFQ (label-free quantification intensities, used for analysis of IAV FP-MS) values were $log_2$-transformed and protein groups only identified by site, reverse matches and potential contaminants excluded from the analysis. Additionally, protein groups quantified by a single peptide or not detected in at least all-but-one replicate of at least one condition were excluded from further analysis. Missing values were replaced by sampling from the normal distribution with the following parameters: 0.3 * standard deviation, mean − 1.8 * standard deviation, where mean and standard deviation relate to the whole AP-MS or FP-MS datasets. For SFV FP-MS data set, statistical comparison was performed using two-sided equal variance $t$-test. For other data sets, generalized linear modeling (core function glm) was used along general linear hypothesis testing (multcomp 1.4 function glht) to quantify protein abundance changes and obtain associated $p$-values. For AP-MS, proteins enriched in particular bait over all controls (log2 fold-change > 2 and FDR-adjusted $p$-value < 0.05) were considered bait interactors. CPSF4 was identified as unique protein satisfying the following conditions: significant interactor of R/SI/G, SI/R, R/FM/G, FM/G but not SI/R or FM/R. For IAV FP-MS, proteins with absolute log2 fold-change > 0.75 and FDR-adjusted $p$-value < 0.01 were considered significantly changing.

For visualization of 1-dimensional density values, R package ggplot2 (version 3.3.2) was used. Specifically, function ggplot and geom_density were used, and for scaled densities the y-axis values were scaled by including y = ..scaled.. into the respective aesthetics (aes) part of the ggplot geom_density function call.

**Quantification of transcript abundance in cell lines by RT-qPCR**
RNA was isolated using MACHEREY-NAGEL NucleoSpin RNA mini kit according to manufacturer instructions. Reverse transcription was performed using Takara PrimeScript RT reagent kit with gDNA eraser according to manufacturer instructions unless otherwise stated. PowerUp SYBR Green (Thermo Fisher, A25778) was used for RT-qPCR on QuantStudio 3 Real-Time PCR system (Thermo Fisher). Ct values, obtained using QuantStudio Design and Analysis Software v1.4.3, were averaged across technical replicates and −ΔCt values as a measure of gene expression were calculated as Ct(RPLP0) − Ct(GOI). −ΔΔCt values as a measure of change in gene expression between conditions were calculated as −ΔCt(KO) − (−ΔCt(NTC)).

PDUI (percentage of distal polyA site usage index, Fig. 1b)[88] as a measure of alternative polyadenylation was calculated as follows:

$$PDUI = Ct(pUTR) - Ct(dUTR) \qquad (1)$$

In this equation, Ct(pUTR) and Ct(dUTR) denote the Ct values of proximal (common) UTR versus distal (elongated) UTR of a particular gene of interest. Note, that higher PDUI values correspond to more polyA site readthrough and thereby on average longer UTRs. To further illustrate this, the equation above can be alternatively expressed as:

$$PDUI = Ct(pUTR) - Ct(dUTR) + Ct(housekeeper) - Ct(housekeeper) \qquad (2)$$

$$PDUI = Ct(housekeeper) - Ct(dUTR) + Ct(pUTR) - Ct(housekeeper) \qquad (3)$$

$$PDUI = (Ct(housekeeper) - Ct(dUTR)) - (Ct(housekeeper) - Ct(pUTR)) \qquad (4)$$

$$PDUI = -\Delta Ct(dUTR) - (-\Delta Ct(pUTR)) \qquad (5)$$

Note, since Ct values are inherently measurements operating in log2 space (i.e. a difference in Ct values of 1 indicates a 2-fold change in target abundance), PDUI and ΔPDUI are also measurements operating in log2 space – for this reason, plots that depict these values also have *(log2)* included in y-axis titles. Naturally, no additional or explicit log2 transformations are required for their calculation, as shown above. PDUI is thereby in principle numerically limited to values below 0, where PDUI = 0 indicates a 100% readthrough of the assayed polyA site, and e.g. PDUI = −2 indicates a 25% readthrough of the assayed polyA site. ΔPDUI is calculated as PDUI(test condition) − PDUI(control condition), and is thereby also a measurement operating in log2 space. ΔPDUI reflects the difference in polyA site readthrough levels between indicated conditions (e.g. ΔPDUI = 1 indicates a 2-fold increase in polyA site readthrough in test condition relative to control condition). The ΔPDUI is positive for 3′UTR elongating events, and negative for 3′UTR shortening events, in respect to control condition.

For detection of APA events, RT-qPCR was performed using primers targeting short (-c) or long (-a) human mRNA isoforms: EIF1-c (AAGGGAGCTTTTGGTGGTAGA, TTTGTGGCAAGCCAGATGTC), EIF1-a (GAAATGGGCAGTGGAGGATTC, CAGTTGACCAGAGAGGGACAA), TFAP2A-c (CCACTGTCCTCCCTTAAAAGC, TACCAGGTGGTCCCAAATGT), TFAP2A-a (CATTTGTTCCTGCGCTCTGA, CGCGTTCGTTTGTCGAGATT), LUC7L3-c (CAGGACTGATGTGACCTACCA, ACTGAGCAAGCCACATGTTT), LUC7L3-a (CTCTGAGCCCTCATCCATTCT, CATGAAGTCCCATCCCACAGA), SFPQ-c (TGGTCTGTTTGGGCAGGTAA, GGAAATTCAGTGGCACAAGGT), SFPQ-a (CTTGGGCAAGCCACTAAGTT, TTACGCACTGCAGAAATGCT), CD47-c (AGTGATGGACTCCGATTTGG, GGGTCTCATAGGTGACAACCA), CD47-a (AAGAGAACTCCAGTGTTGCT, ACGGTAACACAGCTGTAAAACA), and mouse mRNA isoforms: Sfpq-c (TGGCCTGTTTGGGCAGGTAA, GGAAATTCGGTGGCACAAGGT), Sfpq-a (CTGGAGGGGAAATGGGAAAA, AGCATCCTTTCACCACTCTTCA), Eif1-c (TGTTGCGACATGAAGGGTGA, GCACTGGCTCGTACTGAGTT), Eif1-a (CTGTGGCTGTTGGGCTATCT, GGCCACCTTTTAGAGCCAGT), Tfap2a-c (GAGCAATTGCCAAGGGAGAG, TCAGTGGCTGTCTTTACCCA), Tfap2a-a (TGCTACGACGATGTGTGCAT, CGCAACGCAGCAATTGTTTT), Kpnb1-c (GGTACCGCCTCACTCACACA, CCACCTTCACTGTACCGTTG), Kpnb1-a (TTCCACACTACCAGGAAGCA, GGGAACAAGGCACCCTGA).

For transcript quantification, RT-qPCR was performed using the following primers: Cpsf4 (TCATGCACCCTCGATTTGAAC, ACGACCTTTGTAATGGCCGGA), Ifit3 (TGGTCATGTGCCGTTACAGG, GCTGCGAGGTCTTCAGACTT), NS1 (TGACCATGGCCTCTGTACCT, GCATGGAC-CAGTCCCTTGAC), NS1-RBD (TCTTTGGCATGTCCGCAAAC, GTGCTGCCCCTTCCTCTTAG), CPSF4 (CGTGTAAATTCATGCACCCTCG, ACGACCTTTGTAATGGCCGGA), mIL-6 (TAGTCCTTCCTACCCCAATTTCC, TTGGTCCTTAGCCACTCCTTC), M (AGATGAGYCTTCTAACCGA, GCAAAGACATCTTCAAGTYTC), ThoVM (ACCAGTGAGGTCTTTCAGCG, TGTAGGTCCCGGTGGATCTT), RPLP0 (GGATCTGCTGCATCTGCTTG, GCGACCTGGAAGTCCAACTA), U6 (CTCGCTTCGGCAGCACATATAC, GGAACGCTTCACGAATTTGCGTG).

## Protein abundance quantification by Western blotting

At the time of sample harvest, the cells were washed with PBS, lysed in SSB buffer (62.5 mM Tris HCl from 1 M stock solution with pH 6.8, 2% SDS, 10% Glycerol, 50 mM DTT and 0.01% Bromophenol Blue in distilled water) and protein concentrations measured using Pierce 660-nm Protein Assay with an addition of Ionic detergent compatibility kit (Thermo Fischer Scientific) according to manufacturer instructions. Protein concentrations were equalized and up to 10 micrograms of proteins were loaded in NuPAGE Bis-Tris, 1 mm, 4–12% gels (Thermo Fisher Scientific). Protein separation was performed according to gel manufacturer instructions and proteins transferred to 0.22 µm nitro-cellulose membrane (1 h at 100 V in 25 mM Trizma base, 0.192 M Glycine, pH 8.3). The membranes were blocked for 1 h in 5% skim milk or 4% BSA in TBS-T buffer (0.25% Tween-20 in phosphate buffered saline solution) with gentle agitation. For detection of protein abundance by western blotting, the following antibodies were used: GFP (Invitrogen, A6455, 1:1000), HA (Cell Signaling, 2367, 1:1000), HIST3H3 (Abcam, ab1791-100, 1:10,000), CPSF4 (Abcam, ab131218, 1:1000), CPSF3 (Abcam, ab72295, 1:1000), ACTB-HRP (Santa Cruz, sc-47778, 1:5000), TUBB (Santa Cruz, sc-9104, 1:1000), NCL (Thermo Fisher Scientific, 39-6400, 1:1000), msHRP (Cell Signaling, 7076, 1:10,000) and rbHRP (Cell Signaling, 7074, 1:10,000). The antibodies were diluted in 5% skim milk or 4% BSA (TBS-T); the membranes were washed 5× for 5 min with TBS-T between and after incubations with primary and secondary antibodies. Western Lightning ECL Pro (PerkinElmer) was used for band detection according to manufacturer instructions. Normalization of band signals was performed using Image Lab Software (Bio-Rad, version 6.0.1 build 34).

## Quantification of secreted cytokines

For detection of human and mouse IL-6, commercially available ELISA kits were used (mIL-6, BD, 555240 and hIL-6, BD, 555220) according to manufacturer instructions. Fur multiplexed quantification of cytokines from mouse BALF, Bio-Plex Pro Mouse Cytokine 23-plex Assay (Bio-Rad, M60009RDPD) was used according to manufacturer instructions. Statistics were calculated using Student's *t* test.

## In vivo experiments

In order to adhere to the 3R principles and since the CPSF complex is conserved between females and males and there were no indications so far that APA is affected by gender, we opted to only focus on female mice in the herein presented in vivo experiments. 8 weeks-old female C57BL/6J mice were purchased from Janvier Labs and housed under normal conditions (relative humidity 45–65%, temperature 20–24 °C, 12 h light-dark cycle). Mice were anesthetized with 90 mg/kg Ketamine (WDT) and 9 mg/kg Xylazine (Serumwerk Bernburg AG). Mice were inoculated intranasally with $10^5$ pfu of R/SI/G or R/SI/R viruses (8 mice/condition) or mock (4 mice/condition). 3 days post infection, BALF, lung homogenate supernatants and FFPE lungs (5 mice/condition) were harvested for cytokine measurements, virus titrations and histopathology, respectively.

All experiments with mice were carried out in accordance with the guidelines of the Federation for Laboratory Animal Science Associations (FELASA) and the national animal welfare body. Experiments were in compliance with the German animal protection law and were approved by the animal welfare committee of the Regierungspräsidium Freiburg (permits X-18/01 K and G-15/125).

## Reporting summary

Further information on research design is available in the Nature Portfolio Reporting Summary linked to this article.

## Data availability

The mass spectrometry proteomics data have been deposited to the ProteomeXchange Consortium via the PRIDE[89] partner repository with the dataset identifier PXD040456 (https://www.ebi.ac.uk/pride/archive/projects/PXD040456).Sequencing data is publically available via ENA repository (PRJEB54734, https://www.ebi.ac.uk/ena/browser/view/PRJEB54734). Source data are provided with this paper.

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

## Acknowledgements

We want to acknowledge the innate immunity laboratory for critical discussions and suggestions and the Comparative Experimental Pathology (CEP, School of Medicine, TUM) for excellent technical support. Work in the author's laboratories was supported by the Max-Planck Free Floater program, an ERC Consolidator grant (ERC-CoG ProDAP, 817798), the German research foundation (PI 1084/4, PI 1084/5 and TRR179/TP10 and TRR237/A07), KA1-Co-02 "COVIPA" (Helmholtz Association's Initiative and Networking Fund) to A.P. and CNRS, Fondation pour la Recherche Médicale (EQU202003010405) and the European Regional Development Fund (FEDER N∘ EX016008 TARGET-EX and N∘ EX010381 EUROFéRI) to B.R.

## Author contributions

V.B., A.P.: conceptualization. D.S., P.H., C.U., R.D., B.R., K.S., G.K., P.S., A.P.: investigation. N.A.K., T.E.: data analysis. P.A.K., G.K.: resources. G.K., R.R., P.S., A.P.: supervision. G.K., P.S., R.R., A.P.: funding acquisition.

## Funding

## Competing interests

The authors declare no competing interests.

## Additional information

[1]Institute of Virology, TUM School of Medicine, Technical University of Munich, Munich, Germany. [2]Max Planck Institute of Biochemistry, Munich, Germany. [3]Institute of Virology, Medical Center University of Freiburg, Freiburg, Germany. [4]Immunoregulation Laboratory, The Francis Crick Institute, London, UK. [5]Institute of Molecular Oncology and Functional Genomics, TUM School of Medicine, Technical University of Munich, Munich, Germany. [6]Center for Translational Cancer Research (TranslaTUM), TUM School of Medicine, Technical University of Munich, Munich, Germany. [7] Institute for Infectious Diseases, University of Bern, Bern, Switzerland. [8]Institute of Virology and Immunology, Bern & Mittelhäusern, Switzerland. [9]Department of Infectious diseases and Pathobiology, Vetsuisse Faculty, University of Bern, Bern, Switzerland. [10]CNRS, UMR7355 Orleans, France. [11]Experimental and Molecular Immunology and Neurogenetics, University of Orléans, Orléans, France. [12]Institut für allgemeine Pathologie und Pathologische Anatomie, TUM School of Medicine, Technical University of Munich, Munich, Germany. [13]Institute of Molecular Immunology and Experimental Oncology, TUM School of Medicine, Technical University of Munich, Munich, Germany. [14]Faculty of Medicine, University of Freiburg, Freiburg, Germany. [15]Department of Medicine II, TUM School of Medicine, Technical University of Munich, Munich, Germany. [16]German Center for Infection Research (DZIF), Munich Partner Site, Munich, Germany. ✉e-mail: andreas.pichlmair@tum.de

