## [Peer Review File · Nature Communications]

REVIEWER COMMENTS

Reviewer #1 (Remarks to the Author):

mRNA 3'UTR lengthening by alternative polyadenylation attenuates inflammatory responses and correlates with virulence of influenza A virus.

This manuscript by Valter Bergant and co-workers describes how cellular infection with various influenza A (IAV) strains leads to 3'UTR lengthening through alternative poly(A)site (PAS) usage. The authors confirm in their experimental system that this 3'UTR lengthening (or as described in this manuscript, alternative polyadenylation, APA) is caused by the IAV non-structural protein 1 (NS1), and then determine through elegant mutational analysis, which of three previously described motifs in NS1 are required for IAV-induced APA. Contrary to published in vitro interaction studies, the author's mass-spectrometric analysis shows that NS1 amino-acid variants that were previously shown not to interact with CPSF4, precipitate not only CPSF4, but also the WDR33 and FIP1L components of the mammalian polyadenylation specificity factor (mPSF), which is critical for PAS-recognition. By comparing interactors of different NS1 variants, the authors connect NS1's ability to bind CPSF4, with its ability to induce APA. They also identify G184 (motif 3) as critical for association of NS1 with CPSF4, and residues in motif 2 as a fine-tuning mechanism, with F103, M106 accommodating strong interactions to CPSF4 and S103, I106 affording only weak interaction to CPSF4. Using whole cell proteome sampling, the authors conclude that recombinant viruses that bind CPSF4 with high affinity and induce APA (i.e. F103, M106 in motif 2 and G184 in motif 3), can lead to a strong reduction of cellular protein abundance (compared to those that carry F103, M106, R184). In continuation, the authors test the pathogenicity and virulence of viruses with different NS1 variants in mice. They conclude that viruses whose NS1 in transfection experiments leads to reduced APA, can secrete more IL6 and are less pathogenic in mice. This work describes for the first time that all naturally occurring strains tested in this study show APA. The authors conclude their study by analysing a panel of different viruses and describe that some of them induce APA, pinpointing at the central importance of interfering with cellular gene expression to promote viral dominance in the host-pathogen warfare.

This work describes a large body of work that aims to bring clarity into the vast field of NS1-literature. Confounded by the variation of NS1-sequence in different strains, we still lack a complete picture of NS1 actions and therefore this work will provide an important contribution to the IAV field. Most importantly, the finding that all IAV strains tested in this study, including the pandemic H1N1 2009 strain show APA, encourages the authors to speculate that the CPSF4 interaction surface would provide a good target for drug design. Although many of the puzzle pieces that contribute to the conclusions the authors take have been published previously, the authors for the first time combine all of these in one manuscript and extend their observations to a set of different viruses. Whilst it is great to have all these observations in one manuscript, the paper omits critical information in some figure panels, as well as the materials and methods section, which makes it difficult to follow all the conclusions. Overall, I feel that the merit of this publication would not be diminished if some of the data was removed to allow space for a more thorough description of the remaining data (see below in minor comments).

Major comments;

Due to the complexity of NS1 action, but also due to how the data is presented, I find it difficult to fully subscribe to the proposed model of how APA is correlated with protein expression. In Figures 1-4 the authors describe that APA as seen in the PR8 background (R38/S103I106/G184), is not sufficient, to induce notable (significant) changes to expression of cellular proteins. To be able to agree with this claim, it would be important to see a direct comparison between SI and FM variants (as opposed to comparisons between R/SI/G versus R/SI/R, etc.). When looking at gross protein reduction due to host shut-down the comparison should also be drawn to a mock infection. Both these steps should be possible by reanalysing the data.

In contrast, in their in vivo work, the authors look at mL6 secretion, which is induced following viral infection, and whose induction is inhibited by IAV-NS1. In contrast to the claim made in Figure 4 that a R/SI/G NS1 IAV mutant (whose NS1 does induce mild levels of APA) does not induce protein shut-down, when compared to R/SI/R, the authors show in figure 5 and 6 that R/SI/G does lead to reduced IL-6 secretion. This could now be in line with the small changes to protein abundance described in Figure 4C, but I feel again, that an SI versus RM comparison would be helpful, similar to the one performed in Figure 3C, where CPSF4 was identified to bind less well in the R/FM/G. That said, I can see that the authors use the R/SI/R mutant as practically a NS1-dead mutant, to induce ISGs (Interferon Stimulated Genes). However, a UV-inactivated IAV would appear a more appropriate control.

Minor comments:

Figure 1:

- Supp Fig 1a why only considering transcripts that are up?
- The statistics for Supp Figure 1c are not well explained, and the diagram would also benefit from a better explanation. Although to me the comparison to cancer-induced APA is not bringing the paper any benefit, if maintained, it might be useful to depict this as upset plot.
- Fractionation, 8 kb band probably rather 47S pre-rRNA rather than genomic DNA. DNA would remain stuck in the well, which is labelled at a higher position in the gel.
- Figure 1D, where is WSN? Also, cytoplasm receives clearly least APAed transcript. Why?
- Generally, the Log2 transformation makes it difficult to appreciate how low the levels of 3' UTR extended transcripts are being made; (1:50-100).
- Throughout this paper, the authors describe the CPSF4-dependent effect as APA, but do not mention any of the proteins that would be involved in APA, such as for example HuR (ELAV-1). What happens to their abundance?

Figure 2:

- Figure 2a no discussion of other viral proteins used.
- No consistent use of statistics. What is a no-DNA control?
- L190 " and the 184 and its surrounding" would that maybe sound better if rephrased to: "and the amino acid/residue 184."
- Suppl figure 2d all panels are missing log2. Only the top panel is described in the figure legend.
- Suppl Table 4 is suggested to represent all strains sequenced to date. Can the authors please clarify what they mean with this? Alone in 2016 e.g. , 216 NS1 sequences were deposited at NCBI.
- 2e R/FM/G and R/SI/G don't seem to fit into the scheme and are also very lowly expressed. Should they be removed for clarity? Third panel is missing dCT(which gene /NS1?) Log2.

Figure 3

- L 219 RBD-dependent interactors revealed association with..." what sort of association is meant (GO-term/functional association?)
- Figure 3 text. It would be nice to show statistics for suppl Figure 3b, to show that the CPSF4 decrease in G184R is significant, as the double delta graph is difficult to interpret.
- Also – what is the statistical significance for all the other mPSF members? As they seem to change less, is the model that CPSF4 binds independent?

- Figure 3C interesting is that most of mPSF (mammalian polyadenylation specificity factor , including Fip1, CPSF4, WDR33) are precipitated. How significant is their change? Are any of these interactions RNA in-dependent? This should be shown with at least a few of the interaction partners.

Figure 4

- (as discussed above:) It seems that FM/SI determines strength of CPSF4 binding, and R/G determines overall APA. To understand the effect on host-shut down/cellular protein expression, it seems to me that these pairs should be compared. So for the cellular MS, it should be FM/SI G that need to be compared.

- Figure 2f – shows that the difference between A/FM/G and A/FM/R is that R doesn't induce APA anymore. However, Figure 4f third panel shows that between R/FM/G and R/FM/R the latter leads to de-repression of the inhibition – or in other words the G184R mutation in the context of R/FM or A/FM leads to higher expression of proteins belonging to the cell-intrinsic antiviral responses. As both these mutations are also coupled to loss of APA, this would argue that APA is directly connected with expression defects of induced genes.

Figure 6

- 6a) Figure legends does not describe that the numbers describe different MOIs. As this is only barely visible in the fourth panel, this is difficult to understand.
- 6c) figure would be helped by showing what the shRNAs are against.
- IVT4 is not explained .

Figure 7

Some of the viruses are only mentioned as abbreviation.

Materials and methods: CAA needs to be explained when it first appears. In the section for whole cell proteomics.

In conclusion, the experiments presented in this paper have the potential to be very clarifying to the complicated NS1 field. However, given the complexity of NS1 action and preceding work performed with many different strains that have sometimes come to conflicting results, in my eyes a manuscript like this, that has so systematically unpicked the different motifs in NS1 contributing to CPSF4 interaction, would gain a lot of clout if it would describe and discuss less data more extensively.

Reviewer #2 (Remarks to the Author):

The manuscript by Bergant et al uncovers a new mechanism of 3'UTR lengthening by alternative polyadenylation (APA) mediated by the influenza A virus NS1 protein. The authors present some evidence that this process is involved in the immune modulation of cytokine production and inflammation and that this mechanism is also relevant for other pathogenic viruses. Mechanistically, the authors propose that NS1 proteins from various influenza A viruses have either high or low affinity

binding to the CPSF complex. The high affinity binding site of NS1 (F103/M106/G184) to the CPSF complex results in APA, inhibition of host gene expression, and innate immune responses. The low affinity binding site of NS1 (S103/I106/G184) to the CPSF complex induces APA but not a robust inhibition of host gene expression and immunity. The NS1 G184 was shown to be required for induction of APA, and mutation of this residue followed by infection in vivo showed increased levels of cytokines and less pathogenicity. Overall, this is a novel and important observation. However, there are some points that need to be addressed.

1. Introduction: the authors mentioned inhibition of mRNA nuclear export by NS1 without referencing the structural, functional, and genetic approaches demonstrating that this process is mediated by the interaction of NS1 with the main mRNA export receptor NXF1-NXT1 heterodimer (Zhang et al, Nature Microbiology PMID: 31263181). This point needs to be clarified in the Introduction and use the proper reference.

2. Results: In Figure 1, the authors report that APA leads to 3' UTR lengthening of 119 genes and shortening of 18 genes. However, the authors do not discuss enrichment of specific functions or pathways related to these genes/mRNAs. It is unclear the link between the APA hits and the phenotype observed in vivo related to cytokine modulation.

3. In the proteomics study shown in Supplementary Fig. 6, the authors identified differentially expressed proteins in IAV infected cells, but there was no comment on the protein levels of the APA hits. Do the protein levels change for these mRNAs? Additionally, I could not find any comment on what happens to the total levels of the APA hits in the RNA seq dataset.

4. The authors mentioned differences in affinity/efficacy in the interaction between NS1 and the CPSF complex. However, another methodology is required to demonstrate these differences, as affinities were not really measured in the manuscript.

5. The authors have not directly or experimentally linked the mRNAs that undergo APA with regulation of cytokine expression. This needs clarification, as also pointed out in comment #2 above.

6. In Fig. 1d it appears that there is increased nuclear levels of mRNAs that undergo APA compared to the cytoplasmic levels. This appears to be the case for SC35M or SC35M delta NS1. This is not clear in the text. Please clarify.

7. Regarding the point: "RBD-dependent interactors revealed association to dsRNA-binding proteins, which was highly conserved between NS1 proteins with different amino acid residues at positions 103 and 106" (Line 219-220, Results section). Were these interactions detected in the presence of RNase treatment?

8. Minor points: In Supplementary Figures 1a and 2d, some of the colors should be changed for clarity. They are too close in tone.

Reviewer #3 (Remarks to the Author):

Comments:

The authors of this manuscript employed RNA seq, quantitative proteomics, gene transfection, and recombinant influenza A virus (IAV) techniques to investigate IAV-induced alternative polyadenylation (APA) of host transcripts. Their findings indicate that IAV infection leads to extensive host transcript APA that is dependent on the NS1 protein of IAV. It appears that the function of the NS1 on APA depends on its interaction with CPS4, and the interactions are determined by residue 184 in the NS1

and fine-tuned by residues 103 and 106. The authors suggest that the NS1 containing a low affinity epitope for CPS4 (S103/I106/G184) induces APA, while the NS1 containing a high affinity epitope for CPS4 (F103/M106/G184) causes host expression shutoff. They further demonstrate that the G184R mutant strain, which lacks the ability to induce APA of host transcripts, resulted in increased secretion of proinflammatory cytokines but reduced virulence compared to the parental PR8 strain. Finally, they showed that APA of host transcripts is not restricted to IAVs but shared with some other viruses. While the reported results are interesting and important for understanding the mechanisms of IAV infection and virus infection in general, the biological consequences of IAV-induced APA remain uncertain. The authors tried to link the biological effect of IAV-induced APA to changes in the expression of inflammatory cytokines. However, the APA in the IAV-infected cells inhibited the induction of proinflammatory cytokines but resulted in increased severity of the pathological state of mice without causing significant changes in viral load in the lung. The results are not consistent with the conventional view that increased expression of proinflammatory cytokines typically leads to increased severity of the pathological state. In addition, there are concerns about how some of the reported experiments were performed and how data were presented. For example, multiple experiments were performed with only two biological replicates, and it is not clear how statistical analyses were performed on those experiments. In addition, when the authors used RT-qPCR to quantify RNAs in the cells, they used ΔC_t to quantify RNAs instead of the widely accepted $\Delta\Delta C_t$ method. Without using internal controls, ΔC_t values may not correctly represent cellular RNA levels.

Other comments:

1. The authors observed comparable levels of viral transcripts for all strains tested, including the $\Delta NS1$ strain (supp Fig. 2a), despite the fact that replication of $\Delta NS1$ strain is normally severely reduced in IFN-competent cells such as A549. The authors should provide explanations or comments on this.
2. The authors used abbreviations/symbols in many figures without defining them, such as pUTR, dUTR in Fig. 1b, T C N in Fig. 1d, CTR – ThoV M in Fig. 2e legend, SCR in Fig. 3e, M&M in Supp Fig. e legend, ntrf/bkt_ntrf in Supp Table. 5, etc.
3. Fig. 1e and other places: The authors should consider using a Western blot analysis with an antibody against an IAV protein such as NP to monitor replication of the viruses.
4. "NS1 of all influenza A virus strains interact with the CPSF complex" (lane 212) should be changed to "NS1 of all the tested influenza A virus strains interact with the CPSF complex."
5. The authors focused on discussing IL6 despite the expression of multiple cytokines were affected by APA (supp Fig. 4). They should explain why they chose to do so.
6. Why were only female mice used in the studies?
7. The authors should explain how the scaled density and density values were obtained in Fig. 4f and 4g.
8. In Figure 6D, statistical differences were not indicated.
9. Supplementary Figure 1d and 1e: What did the blue bars (after AATAA(+)) indicate?
10. Supplementary Table 4: It may be worth considering the use of a larger IAV database, such as the influenza virus database sponsored by NIH/NIAID, to analyze the NS1 sequences.
11. The authors should provide some basic information about the MS data processing in this manuscript, rather than just mentioning the software name. At the minimum, they should briefly describe what iBAQ and LFQ are.
12. The authors should explain why iBAQ values were used for one experiment and LFQ values were used for another experiment.

Reviewer #1 (Remarks to the Author):

mRNA 3'UTR lengthening by alternative polyadenylation attenuates inflammatory responses and correlates with virulence of influenza A virus.

This manuscript by Valter Bergant and co-workers describes how cellular infection with various influenza A (IAV) strains leads to 3'UTR lengthening through alternative poly(A)site (PAS) usage. The authors confirm in their experimental system that this 3'UTR lengthening (or as described in this manuscript, alternative polyadenylation, APA) is caused by the IAV non-structural protein 1 (NS1), and then determine through elegant mutational analysis, which of three previously described motifs in NS1 are required for IAV-induced APA. Contrary to published in vitro interaction studies, the author's mass-spectrometric analysis shows that NS1 amino-acid variants that were previously shown not to interact with CPSF4, precipitate not only CPSF4, but also the WDR33 and FIP1L components of the mammalian polyadenylation specificity factor (mPSF), which is critical for PAS-recognition. By comparing interactors of different NS1 variants, the authors connect NS1's ability to bind CPSF4, with its ability to induce APA. They also identify G184 (motif 3) as critical for association of NS1 with CPSF4, and residues in motif 2 as a fine-tuning mechanism, with F103, M106 accommodating strong interactions to CPSF4 and S103, I106 affording only weak interaction to CPSF4. Using whole cell proteome sampling, the authors conclude that recombinant viruses that bind CPSF4 with high affinity and induce APA (i.e. F103, M106 in motif 2 and G184 in motif 3), can lead to a strong reduction of cellular protein abundance (compared to those that carry F103, M106, R184). In continuation, the authors test the pathogenicity and virulence of viruses with different NS1 variants in mice. They conclude that viruses whose NS1 in transfection experiments leads to reduced APA, can secrete more IL6 and are less pathogenic in mice. This work describes for the first time that all naturally occurring strains tested in this study show APA. The authors conclude their study by analysing a panel of different viruses and describe that some of them induce APA, pinpointing at the central importance of interfering with cellular gene expression to promote viral dominance in the host-pathogen war-fare.

This work describes a large body of work that aims to bring clarity into the vast field of NS1-literature. Confounded by the variation of NS1-sequence in different strains, we still lack a complete picture of NS1 actions and therefore this work will provide an important contribution to the IAV field. Most importantly, the finding that all IAV strains tested in this study, including the pandemic H1N1 2009 strain show APA, encourages the authors to speculate that the CPSF4 interaction surface would provide a good target for drug design. Although many of the puzzle pieces that contribute to the conclusions the authors take have been published previously, the authors for the first time combine all of these in one manuscript and extend their observations to a set of different viruses. Whilst it is great to have all these observations in one manuscript, the paper omits critical information in some figure panels, as well as the materials and methods section, which makes it difficult to follow all the conclusions. Overall, I feel that the merit of this publication would not be diminished if some of the data was removed to allow space for a more thorough description of the remaining data (see below in minor comments).

- *We thank the reviewer for their time, careful consideration of our work and their recognition of its value.*

Major comments:

Due to the complexity of NS1 action, but also due to how the data is presented, I find it difficult to fully subscribe to the proposed model of how APA is correlated with protein expression. In Figures 1-4 the authors describe that APA as seen in the PR8 background (R38/S103I106/G184), is not sufficient, to induce notable (significant) changes to expression of cellular proteins. To be able to agree with this claim, it would be important to see a direct comparison between SI and FM variants (as opposed to comparisons between R/SI/G versus R/SI/R, etc.). When looking at gross protein reduction due to host shut-down the comparison should also be drawn to a mock infection. Both these steps should be possible by reanalysing the data.

- We thank the reviewer for raising this important point. This is a well-taken point - we have done this analysis and had left out the data for space reasons. As requested, we complemented the existing Fig. 4c with requested comparisons (also highlighted below) and added the respective volcano plots in Supp. Fig. 3d. We additionally added the requested comparisons to figure 4e as comparison of cumulative densities. In these comparisons, global trend towards downregulation of host proteins can be observed for R/FM/G infection (relative to R/SI/G infection) but not for R/FM/R infection (relative to R/SI/R infection). To add additional rigor to this analysis, we additionally provide analogous plots employing different cutoff criteria below. In these column plots, we show that the strong bias towards downregulation of host proteins is present only in X/FM/G containing strains but not R/SI/G (all versus mock, columns 1-6). This is further emphasized by the lack of differences between R/SI/R vs. R/FM/R (column 11), while comparison between R/SI/G vs. R/FM/G again shows downregulation of many host proteins (column 10). This is in line with our previous observations that show G184-dependent downregulation of host protein expression that is apparent when F103/M106 amino acids are present, and absent in the context of S103/I106 (columns 7-9). At this point we would also like to mention that APA can not only modulate mRNA and protein expression, but differentially included UTRs can also serve as scaffolds mediating protein-protein interactions that, for instance, affect protein localization of critical cell-to-cell communication factors (e.g. work by Christine Mayr, PMID: 25896326, 30449617). While we do not know to what extent this happens in our case, it makes follow up of observed APA changes more challenging.*

Revised Fig. 4c

Revised Fig. 4e

Analogous to revised Fig. 4c, but showing numbers of differentially expressed proteins at various cutoffs.

Volcano plots, included in the revised Supp. Fig. 3d.

In contrast, in their in vivo work, the authors look at mL6 secretion, which is induced following viral infection, and whose induction is inhibited by IAV-NS1. In contrast to the claim made in Figure 4 that a R/SI/G NS1 IAV mutant (whose NS1 does induce mild levels of APA) does not induce protein shut-down, when compared to R/SI/R, the authors show in figure 5 and 6 that R/SI/G does lead to reduced IL-6 secretion. This could now be in line with the small changes to protein abundance described in Figure 4C, but I feel again, that an SI versus RM comparison would be helpful, similar to the one performed in Figure 3C, where CPSF4 was identified to bind less well in the R/FM/G. That said, I can see that the authors use the R/SI/R mutant as practically a NS1-dead mutant, to induce ISGs (Interferon Stimulated Genes). However, a UV-inactivated IAV would appear a more appropriate control.

- *We do not fully understand this comment. We feel that reviewer 1 is raising an important point that refers to the multiple functions of the NS1 protein. Essentially, we sense that reviewer 1 would like to see an additional control for the IL6 induction experiments. Here we would like to clarify the rational of our experimental design and why we think that this comparison is most appropriate to address the differential expression of IL6:*
 1. *It is clear that influenza A virus NS1 variants that have high affinity for the CPSF complex profoundly inhibit IL6 expression. This is well documented in literature and*

due to two effects – the general host cell shutoff (mediated by high affinity binding to CPSF) and induction of APA (mediated by low affinity binding to CPSF). We studied IAV-induced APA in contexts where IAV-induced host-shutoff does not occur (Figure 4) since the effects of the two processes would otherwise be impossible to distinguish. For this reason, we focused our functional assays around comparison between NS1 (R/SI/G) (low affinity binder to CPSF, APA inducer) and mutant NS1 (R/SI/R) (no binding to CPSF, no APA). We feel that this is the correct comparison of viral isolates.

2. We feel that using an UV inactivated virus would be very different from using fully replicative viruses. Since the NS1 protein is a non-structural protein, UV inactivated virus would not allow its expression and therefore not recapitulate the multiple characteristics linked to NS1 (i.e. IFN inhibition, dsRNA binding, shutdown, APA). We provide functional evidence using minimal perturbations of the NS1 protein that, to our knowledge, does not have any other function than affecting APA.
3. We do not understand the comment on the *in vivo* experiments. We would like to bring up that repeating *in vivo* experiments for delta NS1 IAV would be difficult from the ethical standpoint, particularly since the function of complete deletion of NS1 from IAV has been studied in detail *in vivo* (e.g. PMID: 17412966). Here we study the effects of NS1 variants that have selective inability to induce APA, which is central to this study.

Minor comments:

Figure1:

- Supp Fig1a why only considering transcripts that are up?

- *We apologize for the unfortunate use of color coding. The analysis was of course not performed only on up-regulated transcripts but in an unbiased manner. The color coding was changed to clarify this.*

- The statistics for Supp Figure 1c are not well explained, and the diagram would also benefit from a better explanation. Although to me the comparison to cancer-induced APA is not bringing the paper any benefit, if maintained, it might be useful to depict this as upset plot.

- *We agree and have amended the figure legend by adding the following statements: “Radar plot further depicts number of overlapping genes found to be APA upon IAV infection and distinct cancer types. Statistics were calculated using one-sided Fisher’s exact test.”*

- Fractionation, 8 kb band probably rather 47S pre-rRNA rather than genomic DNA. DNA would remain stuck in the well, which is labelled at a higher position in the gel.

- *We agree, the panel labeling was changed accordingly.*

- Figure 1D, where is WSN? Also, cytoplasm receives clearly least APAed transcript. Why?

- *WSN strain was not included in this experiment since it followed the same trend as PR/8 and SC35M in initial experiments. Reviewer 1 is right, the proportion of long mRNA isoforms in cytoplasm is generally lower. We do not know the molecular reason for this, but one could imagine that very large RNAs, such as obtained by read-through of poly-A signals are less efficiently exported or that additional nuclear retention signals are more present in the longer 3’UTRs. We added additional comments to the manuscript, lines 162-165.*

- Generally, the Log2 transformation makes it difficult to appreciate how low the levels of 3’UTR extended transcripts are being made; (1:50-100).

- We understand the concern of reviewer 1 and agree that the proportion of long versus short 3'UTR containing mRNA varies across the targets. However, upon infection of A549 cells with IAV, we observed by RT-qPCR that some genes (e.g. TFAP2A, depicted below) exhibit almost 100% extended 3'UTRs. We made similar observations for several other targets, e.g. EIF1A and LUC7L3. However, due to the nature of RT-qPCR based measurements, it is in our opinion best practice to visualize values in log2 space where error distribution is Gaussian, allowing for the use of e.g. t-tests etc. To better explain the plotted values and to allow better appreciation of the magnitude of effects for the reader, we added additional fold-change y-axis in figure 1c as shown below.

Revised Fig. 1c, depicting description of PDUI and Δ PDUI values alongside log2 and linear-equivalent scales.

- Throughout this paper, the authors describe the CPSF4-dependent effect as APA, but do not mention any of the proteins that would be involved in APA, such as for example HuR (ELAV-1). What happens to their abundance?

- We thank the reviewer for raising an important point. We did not detect major impact of IAV infection on abundance of RNA-binding proteins that were previously linked to polyA site selection such as HuR, hnRNPc or many others. We expanded the introduction in lines 98-100 to underline this part. The underlying datasets to backup this statement are provided as supplementary material (including raw data, quanted and quality-filtered abundances, as well as analysed and annotated significance estimates).

Figure 2:

- Figure 2a no discussion of other viral proteins used.

- We thank the reviewer for pointing this out. The manuscript was amended (lines 184-185).

- No consistent use of statistics. What is a no-DNA control?

- Statistics was generally adapted to the nature of the experiment and explanations were added in the revised manuscript. Plasmid transfection experiments were compared to transfection vehicle controls, and infection experiments with IAV mutants were controlled by using respective G184R mutants, although we consistently also show additional mock controls. No-DNA refers to transfection reagent only – we unified this across the Figure 2 and amended the figure legend.

- L190 “ and the 184 and its surrounding” would that maybe sound better if rephrased to: “and the amino acid/residue 184.”

- *We changed this as suggested.*

- Suppl figure 2d all panels are missing log2. Only the top panel is described in the figure legend.

- *We thank the reviewer for pointing this out – figure and figure legend were changed accordingly.*

- Suppl Table 4 is suggested to represent all strains sequenced to date. Can the authors please clarify what they mean with this? Alone in 2016 e.g. , 216 NS1 sequences were deposited at NCBI.

- *We referred to reviewed protein sequences that were deposited in Uniprot and that were provided as supplementary table 4. We thank the reviewer for this comment. In an updated Figure 2d, we now present the amino acid conservation and sequence logos computed across the unique IAV NS1 sequences deposited to NCBI Influenza virus database (35326 sequences) (shown below).*

Revised Fig. 2d – sequence conservation of 35326 NS1 sequences obtained from Influenza virus database and aligned using ClustalOmega algorithm.

- 2e R/FM/G and R/SI/G don't seem to fit into the scheme and are also very lowly expressed. Should they be removed for clarity? Third panel is missing dCT(which gene /NS1?) Log2.

- *While we agree that constructs encoding full length NS1 proteins R/FM/G and R/SI/G are less expressed compared to the effector domain constructs, ΔC_t values around 0 still suggest similar expression to the abundant housekeeping gene RPLP0, which was used for their calculation. Even though this NS1 construct was expressed less efficiently, it was sufficient to induce APA, as expected. The lower expression does thus not influence the interpretation of these findings. For this reasons we would prefer keeping them for the sake of completeness. We also thank the reviewer for pointing out the missing gene label in the third panel – this of course refers to the over-expressed proteins (NS1 or control) and was amended accordingly.*

Figure 3

- L 219 RBD-dependent interactors revealed association with...” what sort of association is meant (GO-term/functional association?)

- *We apologise for the unclear wording, we meant to say that NS1 co-immunoprecipitated numerous dsRNA-binding proteins. This point was clarified in the revised version of the manuscript.*

- Figure 3 text. It would be nice to show statistics for suppl Figure 3b, to show that the CPSF4 decrease in G184R is significant, as the double delta graph is difficult to interpret.

- *We thank the reviewer for this comment. Supp. Fig. 3b was amended as suggested by displaying fdr-adjusted p-values for CPSF4 in both comparisons.*

- Also – what is the statistical significance for all the other mPSF members? As they seem to change less, is the model that CPSF4 binds independent?

- *We thank the reviewer for this comment. We want to place a note of care here - it is in our opinion very difficult to discern “more” or “less” involved interactors using AP-MS data. The enrichment (or p-value) of interaction is influenced by variables like protein solubility, expression levels, expression dynamics, peptide detection rate by the MS, etc.. Due to this, we refrain from making statements about “direct” and “indirect” interaction partners of bait proteins, and cannot comment on the dependent/independent models of the NS1-CPSF4/mPSF complex binding. Specifically referring to Supp. Fig. 3b, we highlighted the other mPSF complex components due to a G184R-dependent trend in enrichment, which is further evident from data presented in Fig. 3c. While these proteins are not significantly enriched when comparing SI/G vs. SI/R, they are highly significant in FM/G vs. FM/R comparison. The underlying data can be appreciated in the different enrichment values in Figure 3c and in Supp. Fig. 3a.*

- Figure 3C interesting is that most of mPSF (mammalian polyadenylation specificity factor, including Fip1, CPSF4, WDR33) are precipitated. How significant is their change? Are any of these interactions RNA in-dependent? This should be shown with at least a few of the interaction partners.

- *We thank the reviewer for raising this point. Indeed, we detected enrichment of most components of mPSF, which we highlight in Fig. 3c. Namely, CPSF2, CPSF3, CPSF4, FIP1L1 and WDR33 are all significantly enriched in the interactome of full length NS1 (both R/SI/G and R/FM/G), thus making a strong point for the interaction of NS1 with the mPSF complex. CPSF4 exhibits the highest enrichment values both for full length NS1 as well as effector domain only NS1 constructs (SI/G and FM/G), which lack the RNA binding domain, emphasizing an RNA-binding-independent nature of the NS1-CPSF4 interaction (Supp. Fig. 3b). As shown in Supp. Fig. 3b, CPSF4 is the only statistically significant hit when comparing effector domains SI/G vs. SI/R, which is the reason it we chose CPSF4 for validation. Co-immunoprecipitation and western blotting in the presence of RNases and DNases confirmed the NS1/CPSF4 interaction, again demonstrating that this interaction is RNA independent. We apologize for unclear description of this experiments – the figure legend was amended.*

Figure 4

- (as discussed above:) It seems that FM/SI determines strength of CPSF4 binding, and R/G determines overall APA. To understand the effect on host-shut down/cellular protein expression, it seems to me that these pairs should be compared. So for the cellular MS, it should be FM/SI G that need to be compared.

- *We thank the reviewer for this comment. We incorporated the suggested changes, alongside major comment above, in the Figure 4c (below, the requested comparisons are highlighted with red rectangle). We find that FM residues lead to a higher number of proteins that are*

less expressed, which is in line with our statement that FM regulates shut-off while G184 regulates APA only.

Revised Fig. 4c.

- Figure 2f – shows that the difference between A/FM/G and A/FM/R is that R doesn't induce APA anymore. However, Figure 4f third panel shows that between R/FM/G and R/FM/R the latter leads to de-repression of the inhibition – or in other words the G184R mutation in the context of R/FM or A/FM leads to higher expression of proteins belonging to the cell-intrinsic antiviral responses. As both these mutations are also coupled to loss of APA, this would argue that APA is directly connected with expression defects of induced genes.

- This is a very interesting point that is actually in line with literature (e.g. PMID: 17442719). Canonical NS1 proteins inhibit interferon induction and function through dsRNA binding (R/XX/X) and host cell shutoff through high affinity CPSF30 binding (X/FM/G combination). Thus, simultaneous mutation of (R>A/FM/G>R) would result in loss of both abilities of canonical NS1 proteins (loss of dsRNA binding and CPSF binding), de-repressing inhibitory functions and explaining the result seen in in Figure 2f. However, from these experiments it is not possible to conclude an involvement of APA since the c-terminal G184 residue would simultaneously alleviate both host cell shutoff and APA.

Figure 6

- 6a) Figure legends does not describe that the numbers describe different MOIs. As this is only barely visible in the fourth panel, this is difficult to understand.

- We thank the reviewer for this comment – we added this information.

- 6c) figure would be helped by showing what the shRNAs are against.

- We thank the reviewer for this comment – the figure legend positions were unified.

- IVT4 is not explained .

- This is immunostimulatory triphosphorylated RNA, and we added an explanation to the Figure 6d legend.

Figure 7

Some of the viruses are only mentioned as abbreviation.

- *Full names were added to the Figure 7a legend.*

Materials and methods: CAA needs to be explained when it first appears. In the section for whole cell proteomics.

- *We went through the manuscript and took care to explain abbreviations upon first use.*

In conclusion, the experiments presented in this paper have the potential to be very clarifying to the complicated NS1 field. However, given the complexity of NS1 action and preceding work performed with many different strains that have sometimes come to conflicting results, in my eyes a manuscript like this, that has so systematically unpicked the different motifs in NS1 contributing to CPSF4 interaction, would gain a lot of clout if it would describe and discuss less data more extensively.

- *We thank the reviewer for this comment. The reviewers provided us with excellent and extensive feedback on how to improve our manuscript and we implemented their suggestions. While we acknowledge that this is a very complicated manuscript and we sympathize with the notion that a streamlined manuscript would be easier to present, a substantially reduced manuscript would lack important information that is required to understand APA in context of IAV. We therefore ask reviewer 1 for his/her understanding that we would like to present the full manuscript as submitted here.*

Reviewer #2 (Remarks to the Author):

The manuscript by Bergant et al uncovers a new mechanism of 3'UTR lengthening by alternative polyadenylation (APA) mediated by the influenza A virus NS1 protein. The authors present some evidence that this process is involved in the immune modulation of cytokine production and inflammation and that this mechanism is also relevant for other pathogenic viruses. Mechanistically, the authors propose that NS1 proteins from various influenza A viruses have either high or low affinity binding to the CPSF complex. The high affinity binding site of NS1 (F103/M106/G184) to the CPSF complex results in APA, inhibition of host gene expression, and innate immune responses. The low affinity binding site of NS1 (S103/I106/G184) to the CPSF complex induces APA but not a robust inhibition of host gene expression and immunity. The NS1 G184 was shown to be required for induction of APA, and mutation of this residue followed by infection in vivo showed increased levels of cytokines and less pathogenicity. Overall, this is a novel and important observation. However, there are some points that need to be addressed.

- *We thank reviewer 2 for her/his comments, for positively evaluating our work and the recognition of its value.*

1. Introduction: the authors mentioned inhibition of mRNA nuclear export by NS1 without referencing the structural, functional, and genetic approaches demonstrating that this process is mediated by the interaction of NS1 with the main mRNA export receptor NXF1-NXT1 heterodimer (Zhang et al, Nature Microbiology PMID: 31263181). This point needs to be clarified in the Introduction and use the proper reference.

- *We thank the reviewer for raising this point. The introduction was amended accordingly (line 71).*

2. Results: In Figure 1, the authors report that APA leads to 3' UTR lengthening of 119 genes and shortening of 18 genes. However, the authors do not discuss enrichment of specific functions or pathways related to these genes/mRNAs. It is unclear the link between the APA hits and the phenotype observed in vivo related to cytokine modulation.

- *We thank the reviewer for raising this point. No statistically significant enrichment of GO-terms (either biological process, molecular function or cellular component) was observed for the set of APA-ed genes (137) taking all genes included in APA analysis (2651) as background. This is not surprising, given the relatively low number of affected genes. We extensively searched for potential motifs that could explain APA sensitivity but yet could not identify signatures that would render a gene APA sensitive or not.*

3. In the proteomics study shown in Supplementary Fig. 6, the authors identified differentially expressed proteins in IAV infected cells, but there was no comment on the protein levels of the APA hits. Do the protein levels change for these mRNAs? Additionally, I could not find any comment on what happens to the total levels of the APA hits in the RNA seq dataset.

- *We thank the reviewer for this comment. We carefully evaluated the abundance of mRNA that are affected in the APA analysis and had presented the results in Supp. Fig. 1a. Comparing conditions where APA happens vs control conditions, we did not find that APA-ed mRNAs exhibit a general trend on their protein expression level (Supp. Table 6). Below, we provide a scatterplot depicting protein abundance changes between R/SI/G versus mock (x-axis) and R/SI/R versus mock. All detected proteins are in grey, products of genes included in APA analysis in black, and products of genes subjected to IAV-induced APA in red. Labels are provided for proteins exhibiting significant difference between infection with R/SI/G versus R/SI/R (adjusted*

p -value < 0.05). We did not observe any clear trends towards up- or down-regulation of APA transcripts.

Scatterplot depicting protein abundance changes in proteomes of A549 cells infected with R/SI/R (y-axis) or R/SI/G (x-axis) versus mock. Grey – all detected proteins; black – products of genes included in the APA analysis (Fig. 1), red – products of genes significantly affected by IAV-induced APA; Gene name labels are provided for proteins with significantly different abundances in A549 cells infected with R/SI/R versus R/SI/G mutant virus.

4. The authors mentioned differences in affinity/efficacy in the interaction between NS1 and the CPSF complex. However, another methodology is required to demonstrate these differences, as affinities were not really measured in the manuscript.

- We agree with reviewer 2, to show affinities we would have to generate recombinant NS1s (and mutants thereof) and recombinant CPSF4 for binding studies. However, we feel that such an analysis would be beyond the scope of this already quite extensive manuscript. We thus changed the wording in the manuscript to “high and low enrichment” (lines 111, 113-115, 117-118, 279-280, 289-290, 295-296, 300, 404-410).

5. The authors have not directly or experimentally linked the mRNAs that undergo APA with regulation of cytokine expression. This needs clarification, as also pointed out in comment #2 above.

- We agree, this point is the subject of expanded discussion in lines 456-459. However, experimentally linking the mRNAs that undergo APA with cytokine dysregulation is challenging for the following reasons:
 1. Despite our deep RNAseq dataset, analyses of APA have limited power in discerning polyA status of transcripts with low mRNA abundance such as many cytokines, transcription factors and kinases (e.g. despite the fact that majority of human genes have multiple polyA sites, of 10757 genes detected in our RNAseq dataset, only 2651 could be evaluated for APA).
 2. There is no clear correlation between APA (either lengthening or shortening) and mRNA or protein abundances (as we show in the scatterplot above and in the manuscript) – the mRNA and protein abundances of APA genes can either be up- or down-regulated, or remain unchanged. While there are some common proteins changing in abundance in the same manner upon infection of cells with different APA-proficient (i.e. R/SI/G,

R/FM/G, A/FM/G) and APA-deficient (i.e. R/SI/R, R/FM/R, A/FM/R), there is no clear connection to immunomodulatory function.

- 3. Beyond protein abundances, APA can also influence protein-protein interactions, mRNA and protein localization, and ultimately the functions of encoded proteins (e.g. work by Christine Mayr, PMID: 30449617).*
- 4. To our knowledge there is no described method to follow up on APA hits – e.g. knock-out screens or transcomplementation assays would not properly recapitulate the APA context of individual genes. We used the depletion of CPSF4 to mimic APA induced by IAV and to recapitulate the effect APA on a global scale in order to be able to clarify functional consequences.*
- 5. The methodology for modulation of polyA site selection on endogenous transcripts in order to screen the activity of individual genes does not exist at the current time.*

For these reasons, we ask the reviewer for understanding that this concern cannot be addressed experimentally within the scope of this manuscript. We provide additional clarification for readers in an expanded discussion (lines 456-459).

6. In Fig. 1d it appears that there is increased nuclear levels of mRNAs that undergo APA compared to the cytoplasmic levels. This appears to be the case for SC35M or SC35M delta NS1. This is not clear in the text. Please clarify.

- We thank the reviewer for raising this point. The differences in polyadenylation status between nuclear and cytoplasmic compartments are also evident in mock-infected samples (Fig. 1d, Supp. Fig. 2d). For that reason, we believe that it reflects a general, yet undescribed biological principle that is unrelated to the infection status of the cells. We added additional comments to the manuscript, lines 162-165.*

7. Regarding the point: "RBD-dependent interactors revealed association to dsRNA-binding proteins, which was highly conserved between NS1 proteins with different amino acid residues at positions 103 and 106" (Line 219-220, Results section). Were these interactions detected in the presence of RNase treatment?

- We did not use RNase treatment prior to AP-MS. RNA constitutes an essential component of NS1 complexes and RNase digestion may therefore affect the "natural" complex formation. However, extensive sonication was performed to prevent precipitation of nucleic acid-bridged interactions. The Materials and methods section was clarified on this point. We inferred NS1 RNA-binding domain dependent interactors according to scheme in Fig. 3a - i.e. by comparing interactors of full length NS1 and NS1 with deletion of RBD (i.e. Effector domain only NS1s).*

8. Minor points: In Supplementary Figures 1a and 2d, some of the colors should be changed for clarity. They are too close in tone.

- Change incorporated for Supp. Fig. 1a.*

Reviewer #3 (Remarks to the Author):

The authors of this manuscript employed RNA seq, quantitative proteomics, gene transfection, and recombinant influenza A virus (IAV) techniques to investigate IAV-induced alternative polyadenylation (APA) of host transcripts. Their findings indicate that IAV infection leads to extensive host transcript APA that is dependent on the NS1 protein of IAV. It appears that the function of the NS1 on APA depends on its interaction with CPS4, and the interactions are determined by residue 184 in the NS1 and fine-tuned by residues 103 and 106. The authors suggest that the NS1 containing a low affinity epitope for CPS4 (S103/I106/G184) induces APA, while the NS1 containing a high affinity epitope for CPS4 (F103/M106/G184) causes host expression shutoff. They further demonstrate that the G184R mutant strain, which lacks the ability to induce APA of host transcripts, resulted in increased secretion of proinflammatory cytokines but reduced virulence compared to the parental PR8 strain. Finally, they showed that APA of host transcripts is not restricted to IAVs but shared with some other viruses.

- *We thank the reviewer for their time, careful consideration of our work and their recognition of its value.*

While the reported results are interesting and important for understanding the mechanisms of IAV infection and virus infection in general, the biological consequences of IAV-induced APA remain uncertain. The authors tried to link the biological effect of IAV-induced APA to changes in the expression of inflammatory cytokines. However, the APA in the IAV-infected cells inhibited the induction of proinflammatory cytokines but resulted in increased severity of the pathological state of mice without causing significant changes in viral load in the lung. The results are not consistent with the conventional view that increased expression of proinflammatory cytokines typically leads to increased severity of the pathological state.

- *We thank the reviewer for highlighting these points. While we detected reduced proinflammatory cytokine secretions in lungs of animals infected with APA-proficient (R/SI/G) relative to APA-deficient (R/SI/R) IAV strains, we also detected a major reduction of secretion of antiviral cytokines such as interferon gamma (IFNG, Fig. 5a). We fully agree that in IAV, overshooting inflammation is associated with increased disease severity. Nevertheless, in the studied context, antiviral proteins, such as interferon gamma, may be decisive factors balancing antiviral and inflammatory cues and at least in part explain the differential mortality.*

In addition, there are concerns about how some of the reported experiments were performed and how data were presented. For example, multiple experiments were performed with only two biological replicates, and it is not clear how statistical analyses were performed on those experiments. In addition, when the authors used RT-qPCR to quantify RNAs in the cells, they used ΔCt to quantify RNAs instead of the widely accepted $\Delta\Delta Ct$ method. Without using internal controls, ΔCt values may not correctly represent cellular RNA levels.

- *We thank the reviewer for raising this point. In the revised manuscript we added additional datapoints for key experiments presented in Figure 2f and Figure 3e. The experiments performed where only two biological replicates were assessed gave qualitative differences, i.e. black or white effects, and were consistent with similar qualitative effects in other cell types or part of kinetic experiments with progressive change of gene expression (e.g. Fig. 1c, Fig. 2f, Fig. 7a). T-tests were performed on (n=2) between indicated samples.*
- *RT-qPCR: We apologize that we were not clear enough in describing ΔCt values in the previous manuscript. As mentioned in material and methods, we used widely accepted housekeeper controls (e.g. RPLP0, ActB) for ΔCt calculations. In the revised version of the manuscript we now additionally added normalization information in figure legends. We feel that visualization of*

Δ Ct values is more informative for the reader as it allows to assess mRNA abundance in relation to a housekeeping gene rather than “relative mRNA abundance” between two conditions. $\Delta\Delta$ Ct (i.e. “fold change”) is particularly problematic when dealing with fold change difference of target RNAs with low or absent basal levels – such as uninfected cells with very low APA or viral transcript levels. The relative abundances can be assessed from Δ Ct values. To facilitate appreciation of the fold change difference of APA in IAV infected cells, we added an additional scale in figure 1c.

Other comments:

1. The authors observed comparable levels of viral transcripts for all strains tested, including the Δ NS1 strain (supp Fig. 2a), despite the fact that replication of Δ NS1 strain is normally severely reduced in IFN-competent cells such as A549. The authors should provide explanations or comments on this.

- We agree with reviewer 3 that the NS1 protein is important to support virus growth in interferon competent systems. We tested for growth of several NS1 deleted IAV strains. In agreement with reviewer 3’s comment, we found that PR8 Δ NS1 is severely affected in virus growth as compared to the PR8 wt strain (see figure below, LC-MS/MS based quantification of viral protein expression upon infection of A549 cells with indicated IAV strains). However, such differences are less accentuated for other strains such as SC35M, where deletion of NS1 is only mildly affecting virus growth in the same in vitro system. To be able to directly compare IAV strains we opted to use SC35M since effects elicited by PR8 and PR8 Δ NS1 could simply be due to large differences in virus growth, as mentioned by reviewer 2.

Heatmap depicting viral protein abundances (relative to mock) in A549 cells, infected with indicated strains of IAV.

2. The authors used abbreviations/symbols in many figures without defining them, such as pUTR, dUTR in Fig. 1b, T C N in Fig. 1d, CTR – ThoV M in Fig. 2e legend, SCR in Fig. 3e, M&M in Supp Fig. e legend, ntrf/bkt_ntrf in Supp Table. 5, etc.

- *We apologize that we missed spelling out abbreviations. We amended the figures, figure legends and table legends accordingly.*

3. Fig. 1e and other places: The authors should consider using a Western blot analysis with an antibody against an IAV protein such as NP to monitor replication of the viruses.

- *We agree that other quantification methods would also be informative, however, we would like to highlight that growth of key recombinant viruses in the infection system used in our study was experimentally addressed previously (PMID: 20926573). We additionally validated this both in vitro using quantitative proteomics (Fig. 4b) and in vivo by titration of lung homogenates (Fig. 5a). We evaluated viral infection rates by RT-qPCR since it was important for this study to quantify APA effects on cellular level in relationship to viral infection levels in the exact same samples (problematic quantitative isolation of both RNA and proteins, limitations of available material).*

4. “NS1 of all influenza A virus strains interact with the CPSF complex” (lane 212) should be changed to “NS1 of all the tested influenza A virus strains interact with the CPSF complex.”

- *We agree, this was changed accordingly.*

5. The authors focused on discussing IL6 despite the expression of multiple cytokines were affected by APA (supp Fig. 4). They should explain why they chose to do so.

- *We thank reviewer 3 for pointing out that the choice of IL-6 was not properly explained. IL-6 was used as well accepted indicator of infection-elicited activation of cellular responses. IL6 is robustly regulated on transcriptional level and can be detected by either RT-qPCR or ELISA with high dynamic ranges, thus allowing for a robust and reliable readout. We clarify the choice of cytokine in the revised version of the manuscript, lines 345-347.*

6. Why were only female mice used in the studies?

- *We agree that sex-differences can affect the outcome of virus infections. However, we do not have any indications that gender differences may affect APA, which is at the center of this study. For welfare reasons and in line with the 3R concept we opted to only do in vivo experiments in females. We ask reviewer 3 for understanding in this matter and will further elaborate on the choice of female mice in material and methods section of the revised manuscript (lines 694-696).*

7. The authors should explain how the scaled density and density values were obtained in Fig. 4f and 4g.

- *We thank reviewer 3 for pointing this out. An additional explanation was added to the materials and methods, lines 629-632: “For visualization of 1-dimensional density values, R package ggplot2 (version 3.3.2) was used. Specifically, the functions ggplot and geom_density were used, and for scaled densities the y-axis values were scaled by including y = ..scaled.. into the respective aesthetics (aes) part of the ggplot geom_density function call.”*

8. In Figure 6D, statistical differences were not indicated.

- We apologize that we did not add the statistical values in the original manuscript. In this particular panel, we initially omitted statistics since we tested for qualitative as opposed to quantitative differences in cytokine expression. As requested, we added statistical analysis for relevant comparisons to the revised version of the manuscript (shown below). All relevant differences are highly significant.

Revised Fig. 6d.

9. Supplementary Figure 1d and 1e: What did the blue bars (after AATAA(+)) indicate?

- The blue bars indicate positions of canonical core polyA sequence AATAAA. We apologize for the lack of description – the figure legend was amended.

10. Supplementary Table 4: It may be worth considering the use of a larger IAV database, such as the influenza virus database sponsored by NIH/NIAID, to analyze the NS1 sequences.

- We thank reviewer 3 for this comment and agree that the wealth of sequencing data deposited in NIH/NIAID could further support analysis of NS1 variations at G184. We now considered all NS1 sequences that were deposited to the Influenza virus database (35326 unique sequences), which gave same results as compared to using all annotated reference sequences from swissprot, which were used for the previous analysis. We updated the respective figure in the revised manuscript.

Revised Fig. 2d.

11. The authors should provide some basic information about the MS data processing in this manuscript, rather than just mentioning the software name. At the minimum, they should briefly describe what iBAQ and LFQ are.

- *We thank the reviewer for this comment and agree that this additional information is very useful for the reader. In the revised version of the manuscript we updated the Materials and methods accordingly (lines 613-615).*

12. The authors should explain why iBAQ values were used for one experiment and LFQ values were used for another experiment.

- *We thank the reviewer for this comment and are happy to expand on this. iBAQ and LFQ intensities are values that represent the protein group's abundance in the measured samples and are obtained from the analysis software MaxQuant. While they are calculated slightly differently, both are widely used in the field to analyse LC-MS/MS experiments such as AP- and FP-MS. We use LFQ values when dealing with larger proteome analysis experiments (such as the comparison between IAV strains presented herein) and iBAQ values when dealing with highly dissimilar samples such as AP-MS experiments (e.g. NS1 AP-MS) or FP-MS experiments with low number of sample groups (e.g. SFV FP-MS). In highly dissimilar samples the normalization algorithm in the LFQ algorithm is, in our experience, in some cases generating inaccurate data for individual proteins. However, either intensity value is commonly used in the MS field and largely subjected to personal preferences of the researchers.*

REVIEWER COMMENTS

Reviewer #1 (Remarks to the Author):

In this revised manuscript Valter Bergant and colleagues have clarified most of my concerns and I can now clearly see their message. I also agree with the authors that the remaining concerns are exceeding the scope of this manuscript. I therefore remain with a few comments:

L101 "directs" should be "direct", I believe.

L 182 a verb is missing, i.e. "we hypothesized that the expression of NS1 may sufficient to reproduce APA in the absence of viral infection" either may be sufficient or may suffice...

Finally, I still have a problem with the presentation of the PDUI Log2. Unless my maths are tricking me, I feel that something is amiss; If the authors are talking about the deltaCt (pUTR-dUTR), then a 100 % readthrough into the APA-UTR extension would correspond to deltaCt=0, of which no Log2 can be formed. If they are talking about the ratio (# molecules pUTR/ # molecules dUTR), then a 100 % readthrough would correspond to 1, of which $\text{Log}_2=1$). However, a readthrough of 25 % would correspond to $\text{Log}_2(4)=2$ and not -2. I.e. if we are talking about ratios on the left axis, then with the negative Log2 values, we are comparing dUTR/pUTR and not as indicated in the figure pUTR/dUTR. Maybe the authors can either correct me or change this accordingly? I agree with the authors that the delta-delta Ct in this case is not making the presentation any clearer.

Reviewer #2 (Remarks to the Author):

The authors addressed my comments. I support publication of this manuscript.

Reviewer #3 (Remarks to the Author):

Comments:

The authors have largely addressed my concerns in the revised manuscript. A few more specific comments:

1. Please provide information on infection doses when describing virus infection-related experiments (e.g., lanes 125-126, 154-155, etc.).
2. Clarify how they performed the affinity purification experiments in lanes 218-221 and Fig. 3. Did they use transfected cells or infected cells?
3. Consider including a supplemental table listing the proteins shown in Fig. 3b. The table should contain important information about the identification of these proteins by LC-MS/MS, such as peptide numbers, spectral counts, and sequence coverage. This will give readers a rough idea of the association between these proteins and the NS1.
4. When describing their hypothesis in lanes 402-405, consider using a different approach to explain the biological effects of different NS1-CPSF associations. For example, a stronger NS1-CPSF interaction leads to APA, while a weaker association results in APA but not shutoff.
5. In lane 182, "the expression of NS1 may sufficient to reproduce APA in the absence of viral infection" should be "the expression of NS1 may be sufficient to reproduce APA in the absence of viral infection".

Reviewer #1 (Remarks to the Author):

In this revised manuscript Valter Bergant and colleagues have clarified most of my concerns and I can now clearly see their message. I also agree with the authors that the remaining concerns are exceeding the scope of this manuscript. I therefore remain with a few comments:

- *We thank the reviewer for additional comments and for positively evaluating our revised manuscript.*

L101 “directs” should be “direct”, I believe.

- *Indeed, was corrected.*

L 182 a verb is missing, i.e. “we hypothesized that the expression of NS1 may sufficient to reproduce APA in the absence of viral infection” either may be sufficient or may suffice...

- *We thank the reviewer for noticing this – was corrected.*

Finally, I still have a problem with the presentation of the PDUI Log2. Unless my maths are tricking me, I feel that something is amiss; if the authors are talking about the deltaCt (pUTR-dUTR), then a 100 % readthrough into the APA-UTR extension would correspond to deltaCt=0, of which no Log2 can be formed. If they are talking about the ratio (# molecules pUTR/ # molecules dUTR), then a 100 % readthrough would correspond to 1, of which Log2=1). However, a readthrough of 25 % would correspond to Log2(4)=2 and not -2. I.e. if we are talking about ratios on the left axis, then with the negative Log2 values, we are comparing dUTR/pUTR and not as indicated in the figure pUTR/dUTR. Maybe the authors can either correct me or change this accordingly? I agree with the authors that the delta-delta Ct in this case is not making the presentation any clearer.

- *We apologise for unclear description of the presented metrics, and thank the reviewer for raising this concern. In order to adhere to the established practices in the field, we chose to use PDUI and ΔPDUI in a similar manner as was defined in Feng et al., TC3A: The Cancer 3'UTR Atlas. Nucleic Acids Res., 2018 (PMID: 30053266). To better illustrate the calculations behind PDUI and ΔPDUI for the readers, we added the following paragraphs to the Materials and Methods (lines 637-658):*

PDUI (percentage of distal polyA site usage index, Fig. 1b) (Feng et al., 2018) as a measure of alternative polyadenylation was calculated as follows:

$$PDUI = Ct(pUTR) - Ct(dUTR)$$

In this equation, Ct(pUTR) and Ct(dUTR) denote the Ct values of proximal (common) UTR versus distal (elongated) UTR of a particular gene of interest. Note, that higher PDUI values correspond to more polyA site readthrough and thereby on average longer UTRs. To further illustrate this, the equation above can be alternatively expressed as:

$$PDUI = Ct(pUTR) - Ct(dUTR) + Ct(\text{housekeeper}) - Ct(\text{housekeeper})$$

$$PDUI = Ct(\text{housekeeper}) - Ct(dUTR) + Ct(pUTR) - Ct(\text{housekeeper})$$

$$PDUI = (Ct(\text{housekeeper}) - Ct(dUTR)) - (Ct(\text{housekeeper}) - Ct(pUTR))$$

$$PDUI = -\Delta Ct(dUTR) - (-\Delta Ct(pUTR))$$

Note, since Ct values are inherently measurements operating in log₂ space (i.e. a difference in Ct values of 1 indicates a 2-fold change in target abundance), PDUI and ΔPDUI are also measurements operating in log₂ space – for this reason, plots that depict these values also have (log₂) included in y-axis titles. Naturally, no additional or explicit log₂ transformations are required for their calculation, as shown above. PDUI is thereby in principle numerically limited to values below 0, where PDUI = 0 indicates a 100% readthrough of the assayed polyA site, and e.g. PDUI = -2 indicates a 25% readthrough of the assayed polyA site. ΔPDUI is calculated as PDUI(test condition) – PDUI(control condition), and is thereby also a measurement operating in log₂ space. ΔPDUI reflects the difference in polyA site readthrough levels between indicated conditions (e.g. ΔPDUI = 1 indicates a 2-fold increase in polyA site readthrough in test condition relative to control condition). The ΔPDUI is positive for 3'UTR elongating events, and negative for 3'UTR shortening events, in respect to control condition.

Reviewer #2 (Remarks to the Author):

The authors addressed my comments. I support publication of this manuscript.

- *We thank the reviewer for positively evaluating our revised manuscript.*

Reviewer #3 (Remarks to the Author):

The authors have largely addressed my concerns in the revised manuscript. A few more specific comments:

- *We thank the reviewer for additional comments and for positively evaluating our revised manuscript.*

1. Please provide information on infection doses when describing virus infection-related experiments (e.g., lanes 125-126, 154-155, etc.).

- *We thank the reviewer for this comment – we added the requested information to figure legends.*

2. Clarify how they performed the affinity purification experiments in lanes 218-221 and Fig. 3. Did they use transfected cells or infected cells?

- *We thank the reviewer for raising this point. The affinity purification experiments were performed using exogenously expressed HA-tagged NS1 constructs (plasmid transfection system). We added clarification to line 220.*

3. Consider including a supplemental table listing the proteins shown in Fig. 3b. The table should contain important information about the identification of these proteins by LC-MS/MS, such as peptide numbers, spectral counts, and sequence coverage. This will give readers a rough idea of the association between these proteins and the NS1.

- *We thank the reviewer for this comment. We would like to highlight that we already provide the quantification and annotation of proteins detected by LC-MS/MS in Supp. Table 4. We additionally provide raw data via PRIDE repository, where we also provide quantification parameters that were used to obtain the published dataset.*

4. When describing their hypothesis in lanes 402-405, consider using a different approach to explain the biological effects of different NS1-CPSF associations. For example, a stronger NS1-CPSF interaction leads to APA, while a weaker association results in APA but not shutoff.

- *We thank the reviewer for this comment and generally agree with proposed example. Nevertheless, we did not directly measure the strength or affinity of interaction between NS1 and the CPSF complex, and thereby decided to use a more conservative wording.*

5. In lane 182, "the expression of NS1 may sufficient to reproduce APA in the absence of viral infection" should be "the expression of NS1 may be sufficient to reproduce APA in the absence of viral infection".

- *We thank the reviewer for noticing this – was corrected.*